# Don't Stop Pretraining? Make Prompt-based Fine-tuning Powerful Learner

**Zhengxiang Shi**
University College London
London, United Kingdom
zhengxiang.shi.19@ucl.ac.uk

**Aldo Lipani**
University College London
London, United Kingdom
aldo.lipani@ucl.ac.uk

## Abstract

Language models (LMs) trained on vast quantities of unlabelled data have greatly advanced the field of natural language processing (NLP). In this study, we re-visit the widely accepted notion in NLP that continued pre-training LMs on task-related texts improves the performance of fine-tuning (FT) in downstream tasks. Through experiments on eight single-sentence tasks and eight sentence-pair tasks in both semi-supervised and fully-supervised settings, we find that conventional continued pre-training does not consistently provide benefits and can even be detrimental for sentence-pair tasks or when prompt-based FT is used. To tackle these issues, we propose Prompt-based Continued Pre-training (PCP), which combines the idea of instruction tuning with conventional continued pre-training. Our approach aims to improve the performance of prompt-based FT by presenting both task-related texts and prompt templates to LMs through unsupervised pre-training objectives before fine-tuning for the target task. Our empirical evaluations on 21 benchmarks demonstrate that the PCP consistently improves the performance of state-of-the-art prompt-based FT approaches (up to 20.1% absolute) in both semi-supervised and fully-supervised settings, even with only hundreds of unlabelled examples. Additionally, prompt-based FT with the PCP outperforms state-of-the-art semi-supervised approaches with greater simplicity, eliminating the need for an iterative process and extra data augmentation. Our further analysis explores the performance lower bound of the PCP and reveals that the advantages of PCP persist across different sizes of models and datasets. Code is available at https://github.com/ZhengxiangShi/PowerfulPromptFT.

## 1 Introduction

Pre-training language models (LMs) [25, 53, 69] over massive unlabelled data and then fine-tuning on task-specific labelled data for the specific downstream task offer large performance gains across NLP tasks. In this study, we re-visit the commonly held belief in NLP [39, 82, 35] that continued pre-training LMs on either task-specific data [1, 56] or in-domain data [54, 97] is generally beneficial for improving the performance of fine-tuning (FT) on downstream tasks. As shown in Figure 1, our experiments on eight single-sentence tasks and eight sentence-pair tasks in both semi- and fully-supervised settings reveal that conventional continued pre-training on task-specific data [35], known as task adaptive pre-training (TAPT) (see

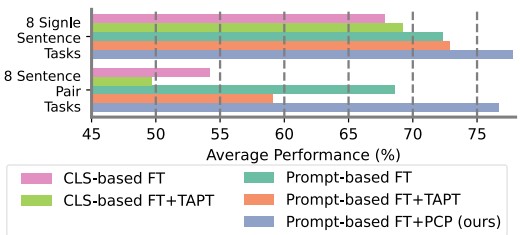

Figure 1: Mean performance of CLS- and prompt-based FT across 16 NLP tasks when trained by *themselves* or in combination with either TAPT or our proposed PCP in the semi-supervised setting. Please refer to Table 1 for details.

37th Conference on Neural Information Processing Systems (NeurIPS 2023).

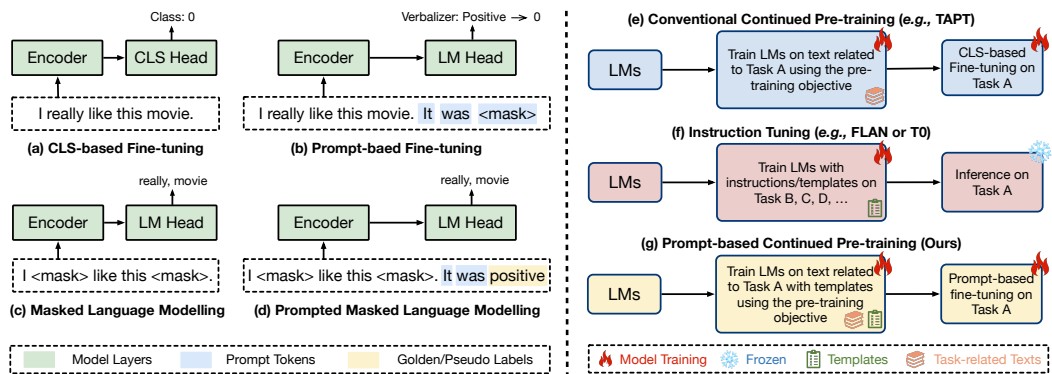

Figure 2: The overview of **Prompt-based Continued Pre-training** (g), in comparison to conventional continued pre-training (e) and instruction tuning (f), along with fine-tuning methods (a,b) and continued pre-training techniques (c,d). The verbalizer functions as a mapping from the task label space to individual words. We use masked language modelling for illustrative purposes, where <mask> represents a masked token in the LM vocabulary.

Figure 2e): (1) can lead to a substantial drop in performance of CLS-based FT (see Figure 2a) on sentence-pair tasks; and (2) may perform unstably across different tasks for prompt-based FT (see Figure 2b), which is typically considered a better alternative to CLS-based FT by previous studies [73, 45] (§4.2). These findings suggest that exclusively **presenting task-related texts to LMs** through continued pre-training may not be the most effective approach for improving the performance of FT in the aforementioned situations.

Recent research [42, 3, 65, 91, 61, 72, 89, 59] on cross-task generalization has demonstrated the impressive improvement on zero-shot or few-shot learning capabilities of LMs (see Figure 2f). These studies suggest that **presenting appropriate instructions/prompt templates to LMs** through training on a range of NLP tasks improves their downstream performance on held-out tasks. Although these works train LMs with different objectives from pre-training phases, we interpret "fine-tuning LMs on a range of NLP tasks" as a special type of continued pre-training. Therefore, we hypothesize that **presenting both task-related texts and instructions/prompt templates to LMs** can relieve the above-mentioned issues for conventional continued pre-training and be beneficial for the target task performance. Rather than improve the generalizability of the LMs with supervised objectives, our work places a greater emphasis on enhancing specific target task performance with unsupervised pre-training objectives.

In this work, we propose Prompt-based Continued Pre-training (PCP) (§3), which integrates instructions/prompt templates into task-related texts with golden or pseudo labels (see Figure 2g). Our experiments demonstrate that PCP consistently improves the performance of state-of-the-art prompt-based FT approaches [28, 100] in both semi- and fully-supervised settings, covering both single sentence tasks and sentence pair tasks, and that the performance gains from PCP exceed those from conventional continued pre-training (TAPT) by a substantial margin (§4.2). In the most favourable case, PCP boosts the performance of prompt-based FT by more than 20% absolute while TAPT results in a 9.2% performance decline. Furthermore, our results show that PCP outperforms state-of-the-art semi-supervised approaches [80, 94, 96, 99, 12] with greater simplicity, eliminating the need for an iterative process and extra data augmentation (§4.3). Additionally, our analysis suggests that the PCP can efficiently improve the performance of prompt-based FT with only hundreds of unlabelled examples. Meanwhile, our analysis explores the performance lower bound of the PCP and reveals that the advantages of PCP persist across different sizes of models and datasets (§4.4). Finally, we outline the limitations of our study and suggest avenues for future research (§6).

In summary, the main contributions of this paper are as follows:

- Our study empirically demonstrates that conventional continued pre-training might not be as effective as initially thought and can even negatively impact fine-tuning performance, particularly in sentence pair tasks or when utilising prompt-based FT;
- Our evaluation on 21 classification and regression NLP tasks shows that our proposed method PCP provides a superior option to conventional continue pre-training for prompt-

based FT. This approach consistently yields performance improvements in diverse model and dataset settings, even with only a few hundred unlabelled examples. Moreover, it can outperform state-of-the-art semi-supervised approaches with greater simplification;

- Our result shows the effectiveness of presenting both task-related texts and templates/instructions to the LMs through unsupervised pre-training objectives on improving the performance of prompt-based FT on downstream tasks. To the best of our knowledge, this is the first work to perform instruction tuning via unsupervised objectives.

## 2 Background

Suppose that we focus on the LMs trained with the masked language modelling (MLM) objective [25, 53]. Let $X = \{x_1, x_2, ..., x_N\}$ be a sequence of tokens, where $N$ represents the total number of tokens. LMs are designed to encode the input text $X$ into a corresponding sequence of hidden vectors $\{\mathbf{h}_i \in \mathbb{R}^d\}$. As shown in Figure 2a, the conventional CLS-based FT [25, 35, 76] trains the output vector $\mathbf{h}$ corresponding to the [CLS] token with an additional head layer (*e.g.,* an MLP layer). However, there is a discrepancy between the pre-training objective (see Figure 2c) and the CLS-based FT objective, which has led to research on prompt-based techniques for better LM performance.

The prompt-based FT is formulated as a MLM problem where the objective is to predict masked tokens [73, 74]. Specifically, the input text $X$ is conditioned with a specific prompt template $\tilde{X} = \mathcal{T}(X)$, which includes one special token [MASK]. The prompt-based FT then maps the output vector associated with the [MASK] token to a label word. The probability of predicting class $y \in \mathcal{Y}$ is computed as:

$$p(y|X) = p(\texttt{[MASK]} = \mathcal{M}(y)|\tilde{X}), \tag{1}$$

where the verbalizer $\mathcal{M} : \mathcal{Y} \to \mathcal{V}$ is a mapping from the task label space to individual words in the vocabulary $\mathcal{V}$.

Prompt-based FT can use either hard or soft prompt templates $\mathcal{T}$, with label words potentially being a part of the prompt templates as well [36, 100]. Hard prompt template [73, 28, 78] requires careful designs of prompts and label words for each task. The use of hard prompts, however, was found to be sub-optimal and sensitive to the choice of the prompt [102, 52]. Soft prompt [52, 100] was then proposed to use unused tokens from the vocabulary $\mathcal{V}$ or additional tokens as tuneable embeddings for prompt template and can be directly trained with the task-specific supervision. This design allows the token embeddings in the prompt template to be updated independently of specific word embeddings after initialization, thus reducing the effort of searching for prompt templates and label words.

## 3 Our Approach: Prompt-based Continued Pre-training (PCP)

In this section, we introduce the proposed method, Prompt-based Continued Pre-training (PCP), which aims to improve the performance of LMs on downstream tasks through continued pre-training with prompt templates, as shown in Figure 2g. Let $L \triangleq \{(X_1, y_1), \dots, (X_n, y_n)\}$ denote $n$ labelled examples and $U \triangleq \{X'_1, \dots, X'_m\}$ denote $m$ unlabelled examples. Our approach consists of two main steps, as described below.

**Step 1: Construct Continued Pre-training Corpus.** Initially, we select a model $F$, pre-trained with the MLM objective and parameterized by $\Theta$. We then train this model using the prompt-based FT, minimizing the target loss function $\ell$ on the labelled examples $L$, as illustrated in Figure 2b:

$$\mathcal{L}(L) = \sum_{X_i, y_i \in L} \ell(y_i, F(\mathcal{T}(X_i), \Theta)), \tag{2}$$

Next, we use the trained model $F$ with the learned parameters $\Theta'$ to generate predictions (termed "pseudo-labels") on the unlabelled samples $U$:

$$y'_i = F(\mathcal{T}(X'_i), \Theta'), \tag{3}$$

For each text example $X$ and its associated (golden or pseudo) label $y$, we create an example for our proposed PCP as $X^{pcp} = \mathcal{T}(X, \mathcal{M}(y))$, where the original [MASK] position is substituted with $\mathcal{M}(y)$. This results in a new corpus, $\mathcal{C} = \{X_i^{pcp}\}_{i=1}^{n+m}$. In the fully-supervised setting, $m = 0$ and all examples use the golden labels.

**Step 2: Perform continued pre-training and prompt-based FT.** We then proceed to further pre-train another model $G$, parameterized by $\Theta$, using the MLM objective on the newly generated corpus $\mathcal{C}$, to obtain the PCP checkpoint $\Phi$ (see Figure 2d). Finally, we train model $G$, initialised by $\Phi$, using Equation 2 with prompt-based FT for downstream tasks.

In comparison to conventional continued pre-training, PCP does not require any modification for the model architecture or training process. The sole difference is the addition of a few extra tokens to the input text during continued pre-training. This modification does not hinder the efficiency of the method, *i.e.,* both conventional continued pre-training and PCP maintain equal levels of efficiency. In this study, we primarily focus on LMs pre-trained with the MLM objective [53]. It is noteworthy to mention that comprehensive exploration of other architectures [25, 70, 14, 65] remains an avenue for future research. Nonetheless, considering prompt-based fine-tuning approaches [52, 47, 51] have already been adapted for different model architectures and pre-training objectives [25, 70, 14, 65]. This implies that extending our method to alternative architectures should be a feasible undertaking.

# 4 Experiments and Results

In this section, we evaluate the proposed method PCP by comparing it with conventional continued pre-training and four state-of-the-art semi-supervised approaches. We assess their relative performance across 21 different classification and regression NLP tasks, including single-sentence and sentence-pair tasks. We conduct additional analysis concerning the performance lower bound of PCP and the effectiveness of the PCP across varying datasets and model sizes.

## 4.1 Experimental Setup

**Datasets.** Our study conducts a comprehensive analysis of 21 NLP datasets, including classification and regression tasks. Following previous studies [28, 36, 100] on prompt-based FT, we derive 8 single-sentence tasks and 8 sentence-pair English tasks from the GLUE benchmark [87], SNLI [13], and 6 other widely used sentence classification tasks (*i.e.,* SST-5, MR, CR, MPQA, Subj, TREC). Additionally, we use 5 popular benchmarks for semi-supervised learning from previous research [34, 21, 94, 48, 29, 77], including IMDB [55], AG NEWS [101], YELP REVIEW[1], YAHOO! ANSWER [18], and AMAZON REVIEW [57]. See dataset details in Appendix §A. We train the model with two different settings: (1) fully-supervised setting, where we train the model with the full training set; and (2) semi-supervised setting, where we sample the same amount of labelled data per class from the full training set. We re-sample the labelled data using the same five seeds for all comparison approaches and report the average performance with an error bar.

**All Comparison Approaches.** In our study, we mainly experiment using the ROBERTA-BASE (125M) and the ROBERTA-LARGE (355M) models. We utilise the conventional CLS-based FT and two state-of-the-art prompt-based FT approaches: (1) "CLS-based FT": fine-tuning with the [CLS] token representation with an extra MLP layer; (2) "Prompt-based FT (hard)": fine-tuning with high-quality manual or auto-generated prompts and label words [73, 28]; and (3) "Prompt-based FT (soft)": fine-tuning with soft prompts using additional tokens for both templates and label words [100]. Since the objective of soft prompt FT is to minimize the reliance on human-designed templates, we unify the template for all tasks here. See the template specifics used for each dataset in Appendix §B. We train these three types of FT approaches from three different types of checkpoints to evaluate their relative effectiveness: (i) the off-the-shelf ROBERTA-LARGE checkpoint; (ii) the task-adaptive pre-training (TAPT) checkpoint [35] (represents the conventional continued pre-training). For sentence pair tasks, we concatenate the two sentences as an input example; and (iii) the proposed PCP checkpoint, obtained in §3. For both (ii) and (iii), we perform MLM on all full training sets except MNLI, MNLI-mm, SNLI, QNLI, and QQP, where we select up to 10k unlabelled examples from the full training sets (see supplementary experiments on the full training sets in Appendix §D). Additionally, we compare the proposed PCP with four state-of-the-art semi-supervised approaches, including FixMatch [80], Dash [96], FlexMatch [99], and AdaMatch [12] (see descriptions of these approaches in Appendix §C), where back-translation [64] is used for data augmentation as previous works [94, 77] and prompt-based FT (hard) is used as the backbone. See hyperparameter and implementation details in Appendix §E.

---

[1] https://www.yelp.com/dataset

**Single Sentence Tasks**

| | SST-2 (acc) | SST-5 (acc) | MR (acc) | CR (acc) | MPQA (acc) | Subj (acc) | TREC (acc) | CoLA (Matt.) |
|---|---|---|---|---|---|---|---|---|
| Majority (full) | 50.9 | 23.1 | 50.0 | 50.0 | 50.0 | 50.0 | 18.8 | 0.0 |
| Prompt-based zero-shot† | 83.6 | 35.0 | 80.8 | 79.5 | 67.6 | 51.4 | 32.0 | 2.0 |
| in-context learning | $84.8_{1.3}$ | $30.6_{0.9}$ | $80.5_{1.7}$ | $87.4_{0.8}$ | $63.8_{2.1}$ | $53.6_{1.0}$ | $26.2_{2.4}$ | $-1.5_{2.4}$ |
| **Fully Supervised Learning** | | | | | | | | |
| CLS-based FT | 95.1 | 59.4 | 90.8 | 90.8 | 89.1 | 96.9 | 96.8 | 54.3 |
| + TAPT | $96.0$ ↑$_{0.9}$ | $60.6$ ↑$_{1.2}$ | $91.4$ ↑$_{0.6}$ | $91.0$ ↑$_{0.2}$ | $89.9$ ↑$_{0.8}$ | $96.9$ ↑$_{0.0}$ | $97.6$ ↑$_{0.8}$ | $43.6$ ↓$_{10.7}$ |
| Prompt-based FT (hard) | 95.2 | 60.0 | 90.8 | 92.4 | 89.4 | 95.9 | 97.8 | 54.7 |
| + TAPT | $93.5$ ↓$_{1.7}$ | $60.4$ ↑$_{0.4}$ | $90.3$ ↓$_{0.5}$ | $90.8$ ↓$_{1.6}$ | $89.5$ ↑$_{0.1}$ | $95.9$ ↑$_{0.0}$ | $97.6$ ↓$_{0.2}$ | $44.0$ ↓$_{10.7}$ |
| + PCP (ours) | $95.5$ ↑$_{0.3}$ | $60.5$ ↑$_{0.5}$ | $91.7$ ↑$_{0.9}$ | $92.8$ ↑$_{0.4}$ | $89.6$ ↑$_{0.2}$ | $96.8$ ↑$_{0.9}$ | $97.8$ ↑$_{0.0}$ | $56.0$ ↑$_{1.3}$ |
| Prompt-based FT (soft) | 94.2 | 59.8 | 90.4 | 92.7 | 87.8 | 96.4 | 97.4 | 61.3 |
| + TAPT | $92.7$ ↓$_{1.5}$ | $59.5$ ↓$_{0.3}$ | $91.8$ ↑$_{1.4}$ | $92.5$ ↑$_{0.2}$ | $89.5$ ↑$_{1.7}$ | $96.8$ ↑$_{0.4}$ | $97.8$ ↑$_{0.4}$ | $52.6$ ↓$_{8.7}$ |
| + PCP (ours) | $94.3$ ↑$_{0.1}$ | $60.7$ ↑$_{0.9}$ | $91.8$ ↑$_{1.4}$ | $92.8$ ↑$_{0.1}$ | $90.4$ ↑$_{2.6}$ | $97.1$ ↑$_{0.7}$ | $98.0$ ↑$_{0.6}$ | $62.0$ ↑$_{0.7}$ |
| **Semi Supervised Learning** | | | | | | | | |
| CLS-based FT | $81.2_{2.7}$ | $41.7_{1.3}$ | $76.3_{3.2}$ | $79.5_{3.8}$ | $65.1_{12.6}$ | $91.7_{0.4}$ | $80.3_{5.8}$ | $26.7_{7.8}$ |
| + TAPT | $88.2_{1.5}$ ↑$_{7.0}$ | $43.4_{2.6}$ ↑$_{1.7}$ | $86.1_{0.7}$ ↑$_{9.8}$ | $86.2_{2.4}$ ↑$_{6.7}$ | $73.7_{4.4}$ ↑$_{8.6}$ | $94.2_{1.5}$ ↑$_{2.5}$ | $80.4_{6.4}$ ↑$_{0.1}$ | $1.9_{2.4}$ ↓$_{24.8}$ |
| Prompt-based FT (hard) | $92.7_{1.3}$ | $46.7_{1.5}$ | $86.2_{1.2}$ | $90.7_{0.8}$ | $80.8_{6.9}$ | $91.0_{1.1}$ | $84.7_{4.4}$ | $7.2_{5.5}$ |
| + TAPT | $92.9_{1.0}$ ↑$_{0.2}$ | $48.9_{1.1}$ ↑$_{2.2}$ | $88.4_{0.5}$ ↑$_{2.2}$ | $89.8_{2.3}$ ↓$_{0.9}$ | $84.6_{4.9}$ ↑$_{3.8}$ | $93.5_{1.1}$ ↑$_{2.5}$ | $85.2_{2.9}$ ↑$_{0.5}$ | $1.4_{3.5}$ ↓$_{5.8}$ |
| + PCP (ours) | $93.6_{0.3}$ ↑$_{0.9}$ | $50.9_{1.3}$ ↑$_{4.2}$ | $89.0_{0.6}$ ↑$_{2.8}$ | $92.3_{0.4}$ ↑$_{1.6}$ | $87.9_{0.5}$ ↑$_{7.1}$ | $95.7_{0.4}$ ↑$_{4.7}$ | $90.6_{3.5}$ ↑$_{5.9}$ | $25.0_{2.9}$ ↑$_{17.8}$ |
| Prompt-based FT (soft) | $92.5_{1.2}$ | $48.0_{0.7}$ | $86.8_{1.4}$ | $90.8_{1.3}$ | $81.2_{6.8}$ | $90.3_{2.1}$ | $83.0_{3.0}$ | $4.9_{3.7}$ |
| + TAPT | $93.4_{0.5}$ ↑$_{0.9}$ | $47.0_{1.2}$ ↓$_{1.0}$ | $88.5_{0.8}$ ↑$_{1.7}$ | $89.6_{3.4}$ ↓$_{1.2}$ | $83.4_{5.1}$ ↑$_{2.2}$ | $93.3_{0.7}$ ↑$_{3.0}$ | $84.5_{2.4}$ ↑$_{1.5}$ | $2.1_{1.8}$ ↓$_{2.8}$ |
| + PCP (ours) | $93.9_{0.3}$ ↑$_{1.4}$ | $50.7_{1.3}$ ↑$_{2.7}$ | $89.8_{0.6}$ ↑$_{3.0}$ | $92.0_{0.5}$ ↑$_{1.2}$ | $88.3_{0.5}$ ↑$_{7.1}$ | $94.9_{0.9}$ ↑$_{4.6}$ | $88.6_{5.4}$ ↑$_{5.6}$ | $21.5_{2.5}$ ↑$_{16.6}$ |

**Sentence Pair Tasks**

| | MNLI (acc) | MNLI-mm (acc) | SNLI (acc) | QNLI (acc) | RTE (acc) | MRPC (F1) | QQP (F1) | STS-B (Pear.) |
|---|---|---|---|---|---|---|---|---|
| Majority (full) | 32.7 | 33.0 | 33.8 | 49.5 | 52.7 | 81.2 | 0.0 | - |
| Prompt-based zero-shot† | 50.8 | 51.7 | 49.5 | 50.8 | 51.3 | 61.9 | 49.7 | -3.2 |
| in-context learning | $52.0_{0.7}$ | $53.4_{0.6}$ | $47.1_{0.6}$ | $53.8_{0.4}$ | $60.4_{1.4}$ | $45.7_{6.0}$ | $36.1_{5.2}$ | $14.3_{2.8}$ |
| **Fully Supervised Learning** | | | | | | | | |
| CLS-based FT | 82.1 | 82.7 | 88.1 | 90.2 | 83.4 | 91.9 | 79.7 | 91.2 |
| + TAPT | $81.0$ ↓$_{1.1}$ | $82.0$ ↓$_{0.7}$ | $86.7$ ↓$_{1.4}$ | $85.6$ ↓$_{4.6}$ | $83.4$ ↑$_{0.0}$ | $91.6$ ↓$_{0.3}$ | $80.2$ ↑$_{0.5}$ | $90.4$ ↓$_{0.8}$ |
| Prompt-based FT (hard) | 85.4 | 85.8 | 89.0 | 89.6 | 88.1 | 93.1 | 73.8 | 91.5 |
| + TAPT | $82.8$ ↓$_{2.6}$ | $83.2$ ↓$_{2.6}$ | $88.3$ ↓$_{0.7}$ | $90.9$ ↑$_{1.3}$ | $83.8$ ↓$_{4.3}$ | $92.7$ ↓$_{0.4}$ | $78.2$ ↑$_{4.4}$ | $91.2$ ↓$_{0.3}$ |
| + PCP (ours) | $86.5$ ↑$_{1.1}$ | $86.2$ ↑$_{0.4}$ | $89.5$ ↑$_{0.5}$ | $91.5$ ↑$_{1.9}$ | $88.5$ ↑$_{0.4}$ | $93.3$ ↑$_{0.2}$ | $79.6$ ↑$_{5.8}$ | $91.9$ ↑$_{0.4}$ |
| Prompt-based FT (soft) | 84.6 | 85.4 | 89.0 | 89.5 | 84.5 | 92.4 | 73.9 | 91.6 |
| + TAPT | $83.5$ ↓$_{1.1}$ | $84.1$ ↓$_{1.3}$ | $88.3$ ↓$_{0.7}$ | $90.9$ ↑$_{1.4}$ | $82.7$ ↓$_{1.8}$ | $92.6$ ↑$_{0.2}$ | $79.9$ ↑$_{6.0}$ | $90.9$ ↓$_{0.7}$ |
| + PCP (ours) | $85.7$ ↑$_{1.1}$ | $86.0$ ↑$_{0.6}$ | $89.5$ ↑$_{0.5}$ | $91.0$ ↑$_{1.5}$ | $85.5$ ↑$_{1.0}$ | $92.6$ ↑$_{0.2}$ | $79.6$ ↑$_{5.7}$ | $91.7$ ↑$_{0.1}$ |
| **Semi Supervised Learning** | | | | | | | | |
| CLS-based FT | $46.2_{0.6}$ | $48.5_{1.0}$ | $45.6_{5.4}$ | $61.4_{8.2}$ | $54.2_{4.3}$ | $73.2_{8.7}$ | $58.5_{3.8}$ | $46.0_{16.3}$ |
| + TAPT | $36.0_{1.0}$ ↓$_{10.2}$ | $36.3_{1.1}$ ↓$_{12.2}$ | $45.7_{3.6}$ ↑$_{0.1}$ | $55.6_{2.7}$ ↓$_{5.8}$ | $53.4_{1.0}$ ↓$_{0.8}$ | $67.7_{8.5}$ ↓$_{5.5}$ | $55.0_{4.1}$ ↓$_{3.5}$ | $48.1_{19.6}$ ↑$_{2.1}$ |
| Prompt-based FT (hard) | $67.3_{1.3}$ | $68.9_{1.2}$ | $76.7_{1.6}$ | $66.5_{4.3}$ | $68.3_{3.1}$ | $75.9_{1.6}$ | $66.8_{1.9}$ | $67.7_{8.1}$ |
| + TAPT | $50.7_{3.9}$ ↓$_{16.6}$ | $52.2_{4.6}$ ↓$_{16.7}$ | $74.5_{3.1}$ ↓$_{2.2}$ | $55.3_{1.1}$ ↓$_{11.2}$ | $59.9_{2.7}$ ↓$_{8.4}$ | $63.2_{6.3}$ ↓$_{12.7}$ | $58.2_{2.6}$ ↓$_{8.6}$ | $63.1_{8.0}$ ↓$_{4.6}$ |
| + PCP (ours) | $75.6_{1.4}$ ↑$_{8.3}$ | $76.8_{0.9}$ ↑$_{7.9}$ | $82.4_{1.3}$ ↑$_{5.7}$ | $85.1_{0.8}$ ↑$_{18.6}$ | $70.2_{2.7}$ ↑$_{1.9}$ | $80.7_{3.3}$ ↑$_{4.8}$ | $71.8_{1.3}$ ↑$_{5.0}$ | $71.5_{8.4}$ ↑$_{3.8}$ |
| Prompt-based FT (soft) | $62.7_{2.2}$ | $65.9_{1.2}$ | $75.4_{0.8}$ | $64.2_{4.7}$ | $68.2_{3.7}$ | $73.0_{10.6}$ | $66.5_{1.8}$ | $63.7_{6.8}$ |
| + TAPT | $46.6_{3.9}$ ↓$_{16.1}$ | $49.5_{6.8}$ ↓$_{16.4}$ | $72.1_{2.0}$ ↓$_{3.3}$ | $55.0_{2.3}$ ↓$_{9.2}$ | $58.4_{2.4}$ ↓$_{9.8}$ | $63.3_{5.8}$ ↓$_{9.7}$ | $58.3_{1.9}$ ↓$_{8.2}$ | $65.3_{6.3}$ ↑$_{1.6}$ |
| + PCP (ours) | $75.4_{0.7}$ ↑$_{12.7}$ | $76.8_{0.3}$ ↑$_{10.9}$ | $82.6_{1.2}$ ↑$_{7.2}$ | $84.3_{2.0}$ ↑$_{20.1}$ | $70.4_{3.2}$ ↑$_{2.2}$ | $80.0_{2.4}$ ↑$_{7.0}$ | $72.3_{1.2}$ ↑$_{5.8}$ | $71.4_{7.8}$ ↑$_{7.7}$ |

***Summary of results: the probability of improving the performance for TAPT and PCP***

| | Single Sentence Tasks | | | | Sentence Pair Tasks | | | |
|---|---|---|---|---|---|---|---|---|
| Checkpoint | TAPT(full) | PCP(full) | TAPT(semi) | PCP(semi) | TAPT(full) | PCP(full) | TAPT(semi) | PCP(semi) |
| CLS-based FT | 87.5 (7/8) | - | 87.5 (7/8) | - | 25.0 (2/8) | - | 25.0 (2/8) | - |
| Prompt-based FT (hard) | 37.5 (3/8) | 100 (8/8) | 75.0 (6/8) | 100 (8/8) | 25.0 (2/8) | 100 (8/8) | 0.0 (0/8) | 100 (8/8) |
| Prompt-based FT (soft) | 50.0 (4/8) | 100 (8/8) | 62.5 (5/8) | 100 (8/8) | 37.5 (3/8) | 100 (8/8) | 12.5 (1/8) | 100 (8/8) |

Table 1: Comparison between the PCP and conventional continued pre-training (TAPT) using RoBERTa-Large. The summary highlights the percentage of positive impact brought by the PCP and TAPT. The mean and standard deviation on test sets are reported over 5 different seeds. In semi-supervised learning, 16 examples per class are used for training, in line with previous studies [28, 52, 100]. Green and red arrows indicate changes with respect to the FT baselines that do not use TAPT or PCP. † represents that no training examples are used. Three extra baselines sourced from [28] are included, where "Majority" refers to the majority class, and "in-context learning" indicates the usage of in-context learning [14] with RoBERTa-Large, without updating any parameters.

## 4.2 Comparison of the PCP and conventional continued pre-training

Table 1 presents and summarises our experimental results on 8 single-sentence tasks and 8 sentence-pair tasks. Below we delve deeper into our two major findings.

**#1. TAPT is not consistently beneficial for sentence pair tasks, nor when prompt-based FT is employed.** Initially, we re-visit the impact of TAPT (representing the conventional continued pre-training) on the CLS-based FT, as shown in Table 1. Our experimental results align with earlier

| Method | IMDB | | AG News | | Yelp Review | | Yahoo! Answer | | Amazon Review | | Mean |
|---|---|---|---|---|---|---|---|---|---|---|---|
| | 20 | 100 | 40 | 200 | 40 | 200 | 40 | 200 | 40 | 200 | |
| DASH [96] | $93.34_{0.7}$ | $93.30_{0.6}$ | $85.00_{2.9}$ | $87.90_{0.3}$ | $47.44_{2.2}$ | $58.85_{1.0}$ | $60.07_{4.7}$ | $66.46_{0.9}$ | $44.09_{2.6}$ | $53.95_{1.0}$ | 69.04 |
| FIXMATCH [80] | $95.26_{0.4}$ | $94.28_{0.5}$ | $85.44_{1.1}$ | $88.21_{0.4}$ | $47.26_{1.2}$ | $58.51_{0.3}$ | $61.56_{6.9}$ | $68.37_{0.7}$ | $44.26_{2.0}$ | $52.33_{1.7}$ | 69.55 |
| FLEXMATCH [99] | $95.22_{0.3}$ | $94.84_{0.4}$ | $85.33_{1.4}$ | $88.57_{0.6}$ | $50.60_{2.5}$ | $58.34_{1.9}$ | $58.09_{3.7}$ | $66.43_{1.3}$ | $45.48_{3.1}$ | $54.19_{1.1}$ | 69.71 |
| ADAMATCH [12] | $95.20_{0.5}$ | $94.94_{0.1}$ | $85.79_{2.1}$ | $88.72_{0.8}$ | $50.42_{2.7}$ | $58.95_{1.9}$ | $63.68_{0.7}$ | $68.09_{0.8}$ | $44.66_{3.4}$ | $53.05_{0.6}$ | 70.35 |
| Prompt-based FT (hard) | $86.78_{2.1}$ | $89.52_{1.6}$ | $84.87_{1.1}$ | $86.99_{0.3}$ | $46.69_{4.2}$ | $58.27_{0.7}$ | $60.63_{1.5}$ | $66.94_{1.1}$ | $44.34_{1.5}$ | $57.01_{0.4}$ | 68.20 |
| + PCP (ours) | $92.49_{1.2}$ | $94.24_{0.9}$ | $87.06_{1.0}$ | $88.94_{0.4}$ | $52.92_{4.5}$ | $63.15_{1.3}$ | $65.58_{1.8}$ | $70.22_{0.9}$ | $53.44_{3.0}$ | $59.64_{1.6}$ | 72.77 |
| Prompt-based FT (soft) | $88.14_{2.9}$ | $90.80_{1.5}$ | $85.65_{1.8}$ | $87.66_{0.3}$ | $45.43_{3.4}$ | $57.12_{1.0}$ | $61.18_{1.5}$ | $67.85_{0.8}$ | $44.52_{3.6}$ | $55.03_{1.5}$ | 68.34 |
| + PCP (ours) | $93.53_{1.5}$ | $94.36_{0.7}$ | $87.26_{0.8}$ | $88.96_{0.6}$ | $50.66_{3.3}$ | $62.92_{1.0}$ | $65.26_{1.5}$ | $70.03_{0.9}$ | $52.78_{3.2}$ | $59.16_{0.8}$ | 72.49 |
| Prompt-based FT (hard)† | 95.60 | | 91.06 | | 68.71 | | 74.30 | | 63.85 | | 78.70 |
| Prompt-based FT (soft)† | 95.50 | | 91.10 | | 69.63 | | 75.66 | | 63.32 | | 79.04 |

Table 2: Comparison between the PCP and four semi-supervised approaches using ROBERTA-LARGE. Each dataset is evaluated with two different labelled data sizes and full training set is used as unlabelled data. † indicates that full training set is used as the labelled data. We report the average Macro-$F_1$ score on the test set across five seeds, with standard deviations as subscripts. For each column, blue represents the best performance and orange stands for the second-best performance.

studies [39, 35, 77], showing that TAPT generally improves the performance of the CLS-based FT on 7 out of 8 single sentence tasks in both semi-supervised and full-supervised setting. However, intriguingly, we observe that TAPT negatively affects the performance of CLS-based FT on 6 out of 8 sentence pair tasks, as summarised in Figure 1 and the at the bottom of Table 1. This finding implies that conventional continued pre-training (TAPT) may not be beneficial for sentence pair tasks.

Moreover, our investigation reveals that TAPT may negatively affect prompt-based FT. Specifically, in the fully supervised setting, TAPT results in reduced performance on 11 out of 16 tasks for prompt-based FT (hard) and on 9 out of 16 tasks for prompt-based FT (soft). In the most favourable scenario, TAPT enhances the performance of prompt-based FT (soft) from 73.9% to 79.9% on the QQP dataset. Conversely, in the least favourable situation, TAPT diminishes the performance of prompt-based FT (hard) from 54.7% to 44.0% on the CoLA dataset. In the semi-supervised setting, TAPT leads to a decline in performance on 12 out of 16 tasks for both prompt-based FT (hard) and prompt-based FT (soft) (see the summary of results in Figure 1 and the at the bottom of Table 1). Particularly, for sentence pair tasks, TAPT results in an average absolute decrease of 9.5% in performance for prompt-based FT. These results suggest that the effectiveness of TAPT varies across different tasks and cannot be universally applied. We conduct additional experiments to confirm the limitations of TAPT persist across different sizes of the pre-training corpus in Appendix D.

**#2. PCP offers consistent and substantial improvements in both semi- and fully-supervised settings.** As depicted in Table 1, our experiments covering 16 datasets in both semi- and fully-supervised settings, including single sentence tasks and sentence pair tasks, reveal that (1) PCP consistently boosts the performance of prompt-based FT; and that (2) the performance gains achieved by PCP consistently exceed those obtained by TAPT by a substantial margin. Specifically, compared to prompt-based FT, PCP leads to more than a 1.0% average absolute improvement in the fully-supervised setting and contributes to an average absolute performance boost of 6.8% in the semi-supervised setting across 16 tasks. Compared to TAPT, PCP yields over a 1.8% average absolute improvement in the fully-supervised setting and contributes to an average absolute performance increase of 11.2% in the semi-supervised setting across 16 tasks. Notably, PCP can produce considerable gains in certain datasets. For instance, it elevates the performance of prompt-based FT (hard) from 7.2% (Matthews Correlation Coefficient) to 25.0%, while TAPT even reduces the performance of the prompt-based FT. Additionally, PCP improves the performance of prompt-based FT (soft) on the QNLI dataset from 64.2% to 84.3% with 31% improvement, while TAPT leads to a 9.2% absolute performance decline. We attribute the improvements to presenting the prompt template to the LMs through the further pre-training phrase, which implies that merely showing task-related texts to the LMs may not be the optimal approach for prompt-based FT.

### 4.3 Comparison of the PCP and state-of-the-art semi-supervised approaches

Table 2 presents our experimental results on five datasets, comparing the proposed PCP with state-of-the-art semi-supervised approaches. Below we delve deeper into our main finding with a discussion.

**The proposed PCP outperforms state-of-the-art semi-supervised approaches on 4 out of 5 tasks.** As shown in Table 2, our proposed PCP approach with either hard or soft variants of prompt-based

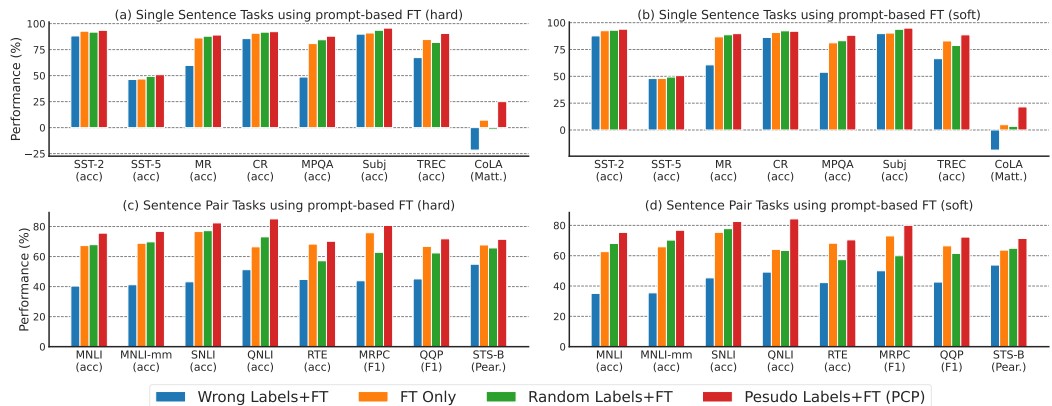

Figure 3: The performance lower bound of the PCP, where "wrong labels" indicates that all labels in the PCP are incorrect and "random labels" indicates that all labels in the PCP are randomly selected. For each dataset, 16 examples per class are used as labelled data and the full training set is used as unlabelled data. The mean performance on test sets is reported over 5 different seeds.

FT outperforms the best-performing semi-supervised approaches on 4 out of 5 datasets. Notably, the prompt-based FT (hard) with the PCP outperforms the best performing semi-supervised approaches (FLEXMATCH) with an absolute 5.5% Macro-$F_1$ score on the AMAZON REVIEW dataset when 200 labelled training examples are used. While the best performing semi-supervised approach, FIXMATCH, outperforms PCP by 1.7% in absolute value on the IMDB dataset using 20 labelled examples, the performance discrepancy narrows as the number of labelled training examples increases. Overall, the prompt-based FT (hard) and (soft) with the PCP outperform all these semi-supervised approaches with an average absolute performance improvement of more than 2% across various datasets and labelled dataset sizes, demonstrating the effectiveness of our proposed approach.

**Discussion.** State-of-the-art semi-supervised approaches typically rely on generating pseudo labels for unlabelled examples in order to train student and teacher models iteratively [4, 15, 95, 27, 94, 29]. However, this iterative process is prone to *confirmation bias* [83, 2, 31], which can result in error accumulation if the pseudo label is incorrect at any iterative step [49, 88, 31, 20]. Various efforts have been made to mitigate *confirmation bias*, such as using only high-confidence pseudo labels [80, 99, 12] or relying heavily on data augmentation [94, 21, 11]. While these efforts make the training process more sophisticated, the issue remains difficult to fully address [20, 77]. Our proposed method offers an alternative way to utilise pseudo labels different from previous semi-supervised approaches [98, 58]. Instead of relying on an iteration process with direct supervision signals from pseudo labels, we incorporate pseudo labels through continued pre-training with an unsupervised objective (*i.e.,* MLM). While our proposed approach may not always outperform semi-supervised approaches across all benchmarks, it delivers highly competitive performance while significantly streamlining the process by removing the necessity for iteration and additional data augmentation. We will discuss the efficiency of the proposed PCP later (§4.4). Additionally, PCP is orthogonal to these semi-supervised approaches and can be combined easily by initialising their backbone from the PCP checkpoint. In future work, we plan to investigate the more specific use cases where our proposed PCP may be preferred over these semi-supervised approaches.

## 4.4 Further Analysis

**#1. What is the lower bound of the model performance using the PCP?** To understand the lower bound of PCP performance, we conduct additional analysis with two different configurations of pseudo labels in PCP: (1) all pseudo labels are incorrect; and (2) all labels are randomly selected. Figure 3 depicts the performance using different types of pseudo labels. We use the prompt-based FT without PCP (shown in yellow) and with PCP (shown in red) as baselines. Experimental results indicate that using incorrect pseudo labels (shown in blue) typically leads to inferior performance. In experiments using two prompt-based FT on 16 datasets, we find that using random labels leads to improved outcomes in 19 out of 32 scenarios. This suggests that PCP with random labels has over a 50% chance of improving the performance of prompt-based FT, indicating that the performance lower

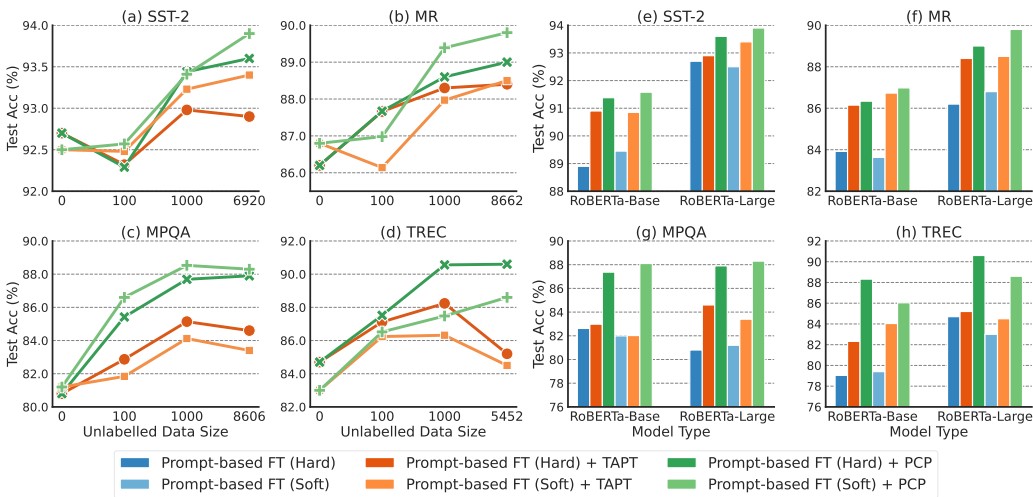

Figure 4: (Left) The effect of different unlabelled data sizes using ROBERTA-LARGE. (Right) The effect of Scaling Laws, where ROBERTA-BASE (123M) and ROBERTA-LARGE (354M). All comparison approaches are trained with 16 examples per class for each dataset.

bound is satisfactory. Additionally, PCP with random labels improves the performance on sentence pair tasks in 8 out of 16 cases, while TAPT leads to poorer results in 15 of 16 cases (refer to Table 1). This suggests that PCP can be advantageous even when using random labels, providing benefits in scenarios where TAPT falls short. Interestingly, unlike prior study [60] on in-context learning [14], where LMs using random labels in demonstrations perform close to those using ground-truth labels, our results show that using pseudo labels assigned by a trained model (shown in red) consistently leads to the better performance, highlighting the importance of accurate pseudo labels.

**#2. What are the requirements of data size and computational resources for the PCP?** To gain a deeper understanding of the efficacy of our proposed PCP method, we conduct additional analysis to determine the number of data points necessary for the PCP. Figure 4 (left) presents the performance of prompt-based FT methods, including both hard and soft variants, across four datasets. The prompt-based FT performance generally improves when the PCP is implemented with more than 1000 unlabelled examples, and some enhancements can be observed even with just 100 unlabelled examples. This indicates that contin-

| Dataset | Size | FT | +TAPT | +PCP |
|---|---|---|---|---|
| IMDB | 23K | $87.3_{1.2}$ | $88.9_{1.3}$ | $91.4_{0.5}$ |
| AG NEWS | 100K | $86.4_{0.9}$ | $87.6_{1.1}$ | $88.0_{0.4}$ |
| YELP REVIEW | 250K | $52.4_{2.5}$ | $60.3_{1.9}$ | $61.44_{2.0}$ |
| AMAZON REVIEW | 250K | $51.2_{1.8}$ | $56.8_{1.2}$ | $57.0_{1.5}$ |
| YAHOO! ANSWER | 500K | $64.9_{0.8}$ | $64.9_{1.1}$ | $69.0_{1.4}$ |

Table 3: Test Results for prompt-based FT (soft) using ROBERTA-BASE with varying continued pre-training corpus sizes. Average Macro-$F_1$ with standard deviations are reported across five seeds. The model is trained on the IMDB dataset using 100 labelled examples and uses 200 labelled examples for other datasets. The best performance for each dataset is highlighted in blue.

ued pre-training (both TAPT and PCP) is not necessarily computationally demanding and can be used efficiently even with only hundreds of training examples. In our experiments, performing the PCP on 1k unlabelled example takes less than 10 minutes using two 24GB NVIDIA 3090 GPUs, and all PCP performance achieved in §4.2 use fewer than 10k unlabelled examples. This is a stark contrast to the previous work [33] that pursued similar objectives (for parameter-efficient fine-tuning) to ours but utilised 10GB of English text data.

**#3. Power of scale.** Our empirical analysis investigates the impact of increasing the backbone LM size on the model performance using the PCP. Figure 4 (right) shows the results of prompt-based FT methods, including hard and soft variants, trained using either TAPT or PCP, on four datasets. The performance of the PCP method largely improves as the backbone LM size expands, which aligns with the scaling laws observed in LMs [41, 37]. Furthermore, the PCP method consistently surpasses other baseline approaches, highlighting the advantages of the PCP across different model sizes.

|  | SST-2 | SST-5 | MR | CR | MPQA | Subj | TREC | CoLA | Mean |
|---|---|---|---|---|---|---|---|---|---|
| Prompt FT | 92.5 | 48.0 | 86.8 | 90.8 | 81.2 | 90.3 | 83.0 | 4.9 | 72.2 |
| Prompt FT +PCP | 93.9 | 50.7 | 89.8 | 92.0 | 88.3 | 94.9 | 88.6 | 21.5 | 77.5 |
| Prompt FT +PCP (Labels Only) | 93.7 | 50.8 | 87.7 | 91.3 | 85.1 | 94.3 | 85.7 | -0.7 | 73.5 |
| Prompt FT +PCP (Template Only) | 90.7 | 43.5 | 88.6 | 92.6 | 82.0 | 95.1 | 84.1 | 0.7 | 72.2 |

Table 4: Ablation study on the inclusion of the template and labels in our proposed PCP. The test Results using soft prompt FT and ROBERTA-LARGE are reported. The best performance for each dataset is highlighted in blue.

|  | SST-2 | SST-5 | MR | CR | MPQA | Subj | TREC | CoLA | Mean |
|---|---|---|---|---|---|---|---|---|---|
| CLS-based FT (1k steps) + TAPT | 88.2 | 43.4 | 86.1 | 86.2 | 73.7 | 94.2 | 80.4 | 1.9 | 69.3 |
| CLS-based FT (5k steps) + TAPT | 89.6 | 43.4 | 86.7 | 87.0 | 72.9 | 94.6 | 79.0 | 1.7 | 69.4 |
| Prompt FT (1k steps) + PCP | 93.9 | 50.7 | 89.8 | 92.0 | 88.3 | 94.9 | 88.6 | 21.5 | 77.5 |

Table 5: Ablation study on the prolonged fine-tuning, where ROBERTA-LARGE is used as the backbone model. The test Results using CLS-based FT and soft prompt FT are reported. The best performance for each dataset is highlighted in blue.

**#4. The impact of a larger continued pre-training corpus on the model performance using PCP and TAPT.** Here we expand our investigation to whether the advantage of the proposed PCP approach persists as the size of the continued pre-training corpus increases. Table 3 presents the performance of prompt-based FT (soft), trained using either TAPT or PCP, across five datasets with varying sizes of unlabelled training examples. These experimental results are consistent with our findings in §4.2 and §4.3, showing that the proposed PCP approach consistently outperforms the model performance using the TAPT even when the larger corpus for continued pre-training is used.

**#5. Ablation study on the label and template inclusion in PCP.** To gain a deeper understanding of the individual contributions of pseudo labels and templates in our proposed PCP method, we conduct an additional ablation study, where we solely utilize pseudo labels or templates. This ablation study is carried out using soft prompt-based fine-tuning. As shown in Table 4, the experimental results reveals that using either labels or templates exclusively will hurt the model's performance compared to our proposed PCP method, highlighting the vital importance of integrating both templates and pseudo labels.

**#6. The impact of prolonged fine-tuning on the model performance.** To ascertain that the effectiveness of our proposed method is not simply due to an extended fine-tuning duration, we conduct additional experiments. We train CLS-based FT 5 times more steps (5k steps in total) from the TAPT checkpoint. As shown in Table 5, our results reveal that prolonged fine-tuning only brings about a marginal improvement of only 0.1% across the eight tasks. Notably, this still falls significantly short of our proposed method (8.1% in absolute).

## 5 Related Work

**Prompt-based Approaches.** In recent years, researchers have been exploring prompt-based approaches to improve the performance of fine-tuning. These approaches can be broadly divided into two research directions. The first direction, known as prompt-based FT, optimizes all parameters in LMs for better performance [73, 28, 52, 100], as discussed in §2. Adaprompt [22] improved the performance of hard prompt-based FT [73, 28] on single sentence tasks through conventional continued pre-training, which is generally consistent with our experimental results. The second direction is parameter-efficient fine-tuning (PEFT) [51, 68, 47, 81, 86], which aims to achieve competitive results while maintaining low computational costs. PPT [33] strives to improve the performance of PEFT [47] by further pre-training the T5 model [70], which pursues a similar idea as ours. However, this method relies on a series of hand-crafted and task-dependent designs for further pre-training, making it less adaptable to novel downstream tasks [86]. Furthermore, it demands a much larger training corpus, as discussed in §4.4. In contrast, our work offers a uniform design across all tasks and focuses on prompt-based FT. In future work, we plan to explore the compatibility of continued pre-training (including both TAPT and our proposed PCP) and PEFT methods.

**Train LMs with Instructions/Templates.** Our work is related to training LMs with templates. Recent studies [42, 3, 65, 91, 61, 72, 89, 59] have explored the idea of LMs training on a variety of NLP tasks with natural language instructions/templates, with the goal of generalizing to unseen tasks. Similar ideas, prompt transfer, have also been explored in the context of PEFT [33, 81, 86, 75], which seeks to learn an effective representation of the soft prompt for the target task by training on other tasks. In our approach, we transfer knowledge from task-related texts with prompt templates that are tailored to a single target task to LMs.

**Semi-supervised Learning.** Our work is related to *semi-supervised learning* [32, 19, 43], with the goal of utilising unlabelled data effectively. Continued pre-training followed by fine-tuning [39, 82, 35] is one type of semi-supervised approaches. While the benefits of continued pre-training are well acknowledged [6, 1, 56], it is commonly assumed that large amounts of data are necessary for continued pre-training [*e.g.,* 50, 38, 33]. Contrarily, our research demonstrates that continued pre-training can improve performance using only a few hundred unlabelled samples. *Self-training* [98, 58] is another powerful semi-supervised approach, which typically uses student-teacher models to assign pseudo-labels to the unlabelled data [46, 44, 83, 62, 4, 15, 27, 94, 80, 29]. Our work offers an alternative way to use pseudo-labels without resorting to an iterative process, as discussed in §4.3.

# 6 Epilogue

**Conclusion.** This study challenges the widely accepted notion in NLP, showing that conventional continued pre-training can be detrimental to model performance, especially for sentence pair tasks and prompt-based FT. As an alternative, we propose Prompt-based Continued Pre-training (PCP), which consistently improves the performance of state-of-the-art prompt-based FT approaches over conventional continued pre-training. Additionally, our proposed PCP outperforms state-of-the-art semi-supervised approaches with a more streamlined process. Further analysis reveals that the advantages of PCP remain consistent across different sizes of models and datasets. This study emphasizes the importance of presenting both task-related texts and templates/instructions to LMs during pre-training for better fine-tuning performance on downstream tasks, contributing to the growing body of research on the optimisation of pre-training and fine-tuning strategies in NLP.

**Limitations and Broader Impact.** We outline several limitations inherent to our research:

- **The scale of language models.** Our experiments utilise relatively modestly-sized language models [53]. The implications of scaling up to more advanced language models, such as the Llama-2 [84] or the mixture-of-experts approach like GPT-4 [63], remains an open question. In the context of large language models, applying PCP with a full set of parameter updates for a specific task may not be justifiable in terms of computational costs. Future research could explore multi-task learning strategies or parameter-efficient continued pretraining.

- **The architecture of language models.** Our work is limited to encoder-only models [25, 53]. To generalize our findings, future research should investigate the effects of our method PCP on encoder-decoder [70] and decoder-only architectures [14].

- **The diversity of tasks.** Our evaluation is confined to text classification and regression tasks. Future research should investigate generative or multi-modal tasks, which may offer more comprehensive insights into the applicability of our methods PCP.

In addition, our work is based on pre-training and prompting methods for LMs. Previous works [8, 14, 7] have extensively discussed the risks and potential harms associated with LMs, including the amplification of undesirable biases learned from unlabelled training data [8, 5, 16]. The energy cost and carbon footprint for our work were approximately 125 kWh and 70 kg $CO_2$e, which are comparatively smaller than LM pre-training [25, 53, 14, 23].

## Acknowledgments and Disclosure of Funding

The authors express their gratitude to the NeurIPS reviewers and area chairs for their insightful discussions. The authors are grateful to Xin Zhao for her contributions to proofreading. Zhengxiang Shi is funded by the Research Studentship from University College London (UCL).

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

## Appendix Overview

The appendix is structured as follows:

**Appendix §A**  provides a brief description for each dataset.

**Appendix §B**  provides details of templates and label words used for each dataset.

**Appendix §C**  presents a brief description of state-of-the-art four semi-supervised (self-training) approaches.

**Appendix §D**  provides the supplementary experimental results to investigate the potential reasons for the ineffectivness of CLS-based fine-tuning on the sentence pair tasks.

**Appendix §E**  provides implementation details and hyperparameters for all comparison methods used in our experiments.

## A  Dataset

In this work, we use 21 popular datasets from previous few-shot learning and semi-supervised learning research.

For experiments in §4.2, we adhere to the approach in [28] and utilise 16 different datasets[2], including SST-2 [79], SST-5 [79], MR [67], CR [40], MPQA [92], Subj [66], TREC [85], CoLA [90], MNLI [93], SNLI [13], QNLI [71], RTE [24, 30, 10], MRPC [26], QQP[3], and STS-B [17]. Consistent with prior research [28], our validation set comprises 16 examples per class from the aforementioned datasets. Additionally, we use 16 examples per class for the training set and the entire training set as the unlabeled set in the semi-supervised setting. We also utilise the full training set for training purposes in the fully supervised setting. For sentence pair tasks, we select at most 10k examples for continued pre-training to reduce the computational costs.

For experiments in §4.3, we follow the setup in [77] and utilise 5 different datasets, including IMDB [55], AG NEWS [101], YELP REVIEW[4], YAHOO! ANSWER [18], and AMAZON REVIEW [57]. Refer to the dataset statistics in Table 6. Our validation set comprises 1,000 examples for each dataset.

## B  Templates for Prompt-based FT

Here we introduce the templates used in two state-of-the-art prompt-based FT approaches for each dataset. For "Prompt-based FT (hard)", we use high-quality manual or auto-generated prompts and label words for each task from previous works [73, 28]. For "Prompt-based FT (soft)", we use the STS-2 template for all single sentence tasks and STS-B template for all sentence pair tasks, while the label words for each task follow the description in Table 7.

## C  ST Frameworks

**FIXMATCH.**  FIXMATCH [80] generates artificial labels using both consistency regularization and pseudo-labelling, where the artificial labels are produced based on weakly-augmented unlabelled data. These artificial labels are then used as targets to train the model on strongly-augmented unlabelled data. FIXMATCH only retains an artificial label if the model assigns a high probability to one of the possible classes.

**DASH.**  DASH [96] extends FIXMATCH by introducing a mechanism with a dynamically adjusted threshold of loss to select a subset of training examples from the unlabelled data for performing SSL.

---

[2]https://github.com/princeton-nlp/LM-BFF/blob/main/data/download_dataset.sh
[3]https://www.quora.com/q/quoradata/
[4]https://www.yelp.com/dataset

#### Single Sentence Tasks

| Dataset | $|\mathcal{Y}|$ | $L$ | #Train | #Test | Type | Labels (classification tasks) |
|---|---|---|---|---|---|---|
| SST-2 | 2 | 19 | 6,920 | 872 | Sentiment | positive, negative |
| SST-5 | 5 | 18 | 8,544 | 2,210 | Sentiment | v. pos., positive, neutral, negative, v. neg. |
| MR | 2 | 20 | 8,662 | 2,000 | Sentiment | positive, negative |
| CR | 2 | 19 | 1,775 | 2,000 | Sentiment | positive, negative |
| MPQA | 2 | 3 | 8,606 | 2,000 | Opinion Polarity | positive, negative |
| Subj | 2 | 23 | 8,000 | 2,000 | Subjectivity | subjective, objective |
| TREC | 6 | 10 | 5,452 | 500 | Question cls. | abbr., entity, description, human, loc., num. |
| CoLA | 2 | 8 | 8,551 | 1,042 | Acceptability | grammatical, not_grammatical |
| IMDB | 2 | 149 | 8,000 | 1,000 | Movie Review | positive, negative |
| AG NEWS | 2 | 37 | 8,000 | 1,000 | News Topic | world, sports, business, sci/tech |
| YELP REVIEW | 2 | 134 | 8,000 | 1,000 | Review Sentiment | 1, 2, 3, 4, 5 |
| AMAZON REVIEW | 2 | 79 | 8,000 | 1,000 | Review Sentiment | 1, 2, 3, 4, 5 |
| YAHOO! ANSWER | 2 | 32 | 8,000 | 1,000 | Topic Classification | culture, science, health, education, computer, sports, business, music, family, politics |

#### Sentence Pair Tasks

| Dataset | $|\mathcal{Y}|$ | $L$ | #Train | #Test | Type | Labels (classification tasks) |
|---|---|---|---|---|---|---|
| MNLI | 3 | 22/11 | 392,702 | 9,815 | NLI | entailment, neutral, contradiction |
| SNLI | 3 | 14/8 | 549,367 | 9,842 | NLI | entailment, neutral, contradiction |
| QNLI | 2 | 11/30 | 104,743 | 5,463 | NLI | entailment, not_entailment |
| RTE | 2 | 49/10 | 2,490 | 277 | NLI | entailment, not_entailment |
| MRPC | 2 | 22/21 | 3,668 | 408 | Paraphrase | equivalent, not_equivalent |
| QQP | 2 | 12/12 | 363,846 | 40,431 | Paraphrase | equivalent, not_equivalent |
| STS-B | $\mathcal{R}$ | 11/11 | 5,749 | 1,500 | Sent. Similarity | - |

Table 6: The datasets evaluated in this work. $|\mathcal{Y}|$: # of classes for classification tasks (with one exception: STS-B is a real-valued regression task over the interval $[0, 5]$). $L$: average # of words in input sentence(s). Note that we only sample examples from the original training set in our few-shot experiments.

#### Single Sentence Tasks

| Task | Template | Label words |
|---|---|---|
| SST-2 | $<S_1>$ It was [MASK] . | positive: great, negative: terrible |
| SST-5 | $<S_1>$ It was [MASK] . | v.positive: great, positive: good, neutral: okay, negative: bad, v.negative: terrible |
| MR | $<S_1>$ It was [MASK] . | positive: great, negative: terrible |
| CR | $<S_1>$ It was [MASK] . | positive: great, negative: terrible |
| MPQA | $<S_1>$ is [MASK] . | positive: positive, negative: negative |
| Subj | $<S_1>$ This is [MASK] . | subjective: subjective, objective: objective |
| TREC | [MASK] : $<S_1>$ | abbreviation: Expression, entity: Entity, description: Description human: Human, location: Location, numeric: Number |
| COLA | $<S_1>$ This is [MASK] . | grammatical: correct, not_grammatical: incorrect |
| IMDB | $<S_1>$ It was [MASK] . | positive: great, negative: terrible |
| AG NEWS | $<S_1>$ It was [MASK] . | World: world, Sports:sports, Business: business, Sci/Tech: tech |
| YELP REVIEW | $<S_1>$ It was [MASK] . | 0: 0, 1: 1, 2: 2, 3: 3, 4: 4 |
| AMAZON REVIEW | $<S_1>$ It was [MASK] . | 0: 0, 1: 1, 2: 2, 3: 3, 4: 4 |
| YAHOO! ANSWER | $<S_1>$ It was [MASK] . | culture: culture, science: science, health: health, education: education computer: computer, sports: sports, business: business music: music, family: family, politics: politics |

#### Sentence Pair Tasks

| Task | Template | Label words |
|---|---|---|
| MNLI | $<S_1>$ ? [MASK] , $<S_2>$ | entailment: Yes, neutral: Maybe, contradiction: No |
| SNLI | $<S_1>$ ? [MASK] , in this case $<S_2>$ | entailment: Yes, neutral: Maybe, contradiction: No |
| QNLI | $<S_1>$ ? [MASK] , $<S_2>$ | entailment: Yes, not_entailment: No |
| RTE | $<S_1>$ ? [MASK] , I think that $<S_2>$ | entailment: Clearly, not_entailment: Yet |
| MRPC | $<S_1>$ [MASK] , $<S_2>$ | equivalent: Yes, not_equivalent: No |
| QQP | $<S_1>$ [MASK] , $<S_2>$ | equivalent: Yes, not_equivalent: No |
| STS-B | $<S_1>$ [MASK] , $<S_2>$ | $y_u$: Yes, $y_l$: No |

Table 7: Templates and label words used for "Prompt-based FT (hard)". We use the STS-2 and STS-B template for all single sentence tasks and sentence pair tasks using "Prompt-based FT (soft)", respectively.

**FLEXMATCH.** FLEXMATCH [99] also extends FIXMATCH by introducing the concept of curriculum learning [9] to flexibly adjust thresholds for different classes at each time step and select unlabelled data and their pseudo labels that are more likely to be informative.

**ADAMATCH.** ADAMATCH [12] aims to solve domain adaptation problems in SSL and build a high-accuracy model that trains on and tests on different data distributions. ADAMATCH builds on FIXMATCH and introduces a relative confidence threshold and a modified distribution alignment from [11].

## D Supplementary Experiment

In this section, we investigate why TAPT does not work on sentence pair tasks. We have evaluated three possible explanations for TAPT's ineffectiveness on sentence pair tasks: (1) **dataset size for continued pre-training**, (2) **sentence pairs with higher similarity than what was observed in pre-training data**, and (3) **lack of separation within sentence pairs**. Our experimental results suggest that the ineffectiveness of TAPT on the sentence pair tasks is not an isolated incident but a recurring issue. Below we discuss each setting in detail.

**#1. The impact of continued pre-training (TAPT) with a larger pre-training corpus on the performance of the prompt-based FT on sentence pair tasks.** In Section 4.2, we randomly selected at most 10k unlabeled examples from the full training sets of MNLI, MNLI-mm, SNLI, QNLI, and QQP, as corpus for continued pre-training due to our limited academic computational resources. For all other tasks, we use the full training set for continued pre-training because there are fewer than 10k examples in their training sets. To verify our findings that "TAPT *is not consistently beneficial for sentence pair tasks, nor when prompt-based* FT *is employed*" holds true when utilising larger continued pre-training corpus, we perform conventional continued pre-training (TAPT) on the full training set on MNLI, MNLI-mm, SNLI, QNLI, and QQP.

| Dataset | MNLI | MNLI-mm | SNLI | QNLI | QQP |
|---|---|---|---|---|---|
| Corpus Size | 393k | 393k | 549k | 104k | 364k |
| CLS-based FT | $46.2_{0.6}$ | $48.5_{1.0}$ | $45.6_{5.4}$ | $61.4_{8.2}$ | $58.5_{3.8}$ |
| + TAPT | $34.7_{0.4}\downarrow$ | $35.1_{0.6}\downarrow$ | $41.8_{2.7}\downarrow$ | $54.8_{2.0}\downarrow$ | $62.6_{2.9}\uparrow$ |
| Prompt-based FT (hard) | $67.3_{1.3}$ | $68.9_{1.2}$ | $76.7_{1.6}$ | $66.5_{4.3}$ | $66.8_{1.9}$ |
| + TAPT | $47.8_{5.6}\downarrow$ | $47.9_{5.2}\downarrow$ | $47.5_{9.4}\downarrow$ | $53.5_{0.8}\downarrow$ | $53.5_{0.8}\downarrow$ |
| Prompt-based FT (soft) | $62.7_{2.2}$ | $65.9_{1.2}$ | $75.4_{0.8}$ | $64.2_{4.7}$ | $66.5_{1.8}$ |
| + TAPT | $45.4_{3.7}\downarrow$ | $45.8_{4.1}\downarrow$ | $50.2_{3.9}\downarrow$ | $53.8_{0.9}\downarrow$ | $53.8_{0.9}\downarrow$ |

Table 8: Test Results using ROBERTA-LARGE, with corresponding continued pre-training corpus sizes for each task. The mean performance with standard deviations are reported across five seeds.

Table 8 presents the performance of the CLS-based FT, prompt-based FT (hard), and prompt-based FT (soft) using the TAPT. The experimental results reveal that the TAPT generally results in poorer performance, even when a larger continued pre-training corpus is used. Notably, the performance of these fine-tuning approaches could be even worse than those achieved using a smaller continued pre-training corpus (refer to results in Table 1), suggesting that training with a larger corpus is not an effective solution to the issues of conventional continued pre-training (TAPT).

| | MNLI | MNLI-mm | SNLI | QNLI | RTE | MRPC | QQP | STS-B | Mean |
|---|---|---|---|---|---|---|---|---|---|
| CLS-based FT | 46.2 | 48.5 | 45.6 | 61.4 | 54.2 | 73.2 | 58.5 | 46.0 | 54.2 |
| +TAPT | 36.0 | 36.3 | 45.7 | 55.6 | 53.4 | 67.7 | 55.0 | 48.1 | 49.7 |
| +TAPT (Tokenizer Sep) | 36.4 | 37.5 | 50.5 | 58.8 | 50.8 | 63.5 | 59.2 | 48.8 | 50.7 |
| +TAPT (PCP Sep) | 36.3 | 36.7 | 64.6 | 58.3 | 51.2 | 65.3 | 57.4 | 44.5 | 51.8 |
| +TAPT (random sent pair) | 34.8 | 35.4 | 37.7 | 52.2 | 51.2 | 64.8 | 56.9 | 23.8 | 44.6 |
| +TAPT (first sent only) | 35.6 | 35.9 | 42.7 | 52.2 | 52.6 | 62.5 | 53.6 | 16.7 | 44.0 |

Table 9: Ablation study on the performance of CLS-based fine-tuning with different settings of conventional continued pre-training, where ROBERTA-LARGE is used as the backbone model.

**#2. High similarity within sentence pairs.** We consider that the high similarity between the sentence pairs might conflict with the word distribution that the model has observed during model pre-training. For instance, in the MNLI task, two sentences are *Salt kept the town fed* and *Salt kept the*

*town thriving*. To explore this, we perform TAPT on two different settings, one where we continually pre-train TAPT on randomly paired sentences within the dataset and another where we continually pre-train TAPT using just the first sentence of each pair. As shown in Table 9, the experimental results show that training TAPT with either case leads to even worse performance.

**#3. Token-based separation of sentence pairs.** In an attempt to mitigate the effect above, we also consider that distinguishing two sentences using distinct tokens might make a difference. To test this, we perform TAPT with two types of separate tokens, the special token from the tokenizer and the template used in PCP (without labels). As shown in Table 9, training TAPT with separate tokens between two sentences can somewhat mitigate the performance drop for CLS-based fine-tuning on the sentence pair tasks. However, the results remain inferior compared to CLS-based fine-tuning without the use of TAPT.

In conclusion, our investigations highlight the difficulties that TAPT faces on sentence pair tasks, while our proposed method PCP provides a simple yet effective solution. We hypothesize that TAPT's ineffectiveness for CLS-based fine-tuning on sentence pair tasks might be due to various factors, which we leave for a more comprehensive investigation in future work.

# E    Implementation Details

Our code is implemented using Pytorch[5] and Huggingface[6]. The semi-supervised approaches are implemented upon the repository[7]. Below, we provide a comprehensive list of the hyperparameters used in our code. For fine-tuning, as shown in Table 10, we conduct a grid search for learning rates within the set {1e-5, 2e-5, 5e-5}, and choose a batch size of 8. In each trial, we train the model for 1,000 steps, evaluate performance every 100 steps, and select the best checkpoint based on optimal performance on the evaluation set. The best performance is determined by the relevant evaluation metric. For continued pre-training, we utilise the same set of hyperparameters for both TAPT and PCP, as shown in Table 11. The learning rate and unlabeled data size are closely linked and need to be adjusted simultaneously. As a general guideline, we suggest decreasing the learning rate as the unlabeled data size decreases. In contrast to its predecessor, BERT [25], which uses the next sentence prediction objective, ROBERTA [53] is trained solely with the masked language model (MLM) objective, specifically the cross-entropy loss on predicting randomly masked tokens. RoBERTa dynamically alters the masking pattern applied to training examples, typically employing a masking probability of 0.15. Additionally, Table 12 lists the hyperparameters for self-training approaches, where a grid search for learning rates within the set {1e5, 2e-5, 5e-5} is conducted.

---

[5]https://pytorch.org/
[6]https://huggingface.co/
[7]https://github.com/amzn/pretraining-or-self-training

| Hyperparameter | Assignment |
| --- | --- |
| number of steps | 1000 steps (evaluate every 100 steps) |
| batch size | 8 |
| maximum learning rate | 1e-05, 2e-5, 5e-5 |
| maximum sequence length | 128, 256 |
| learning rate optimizer | AdamW |
| Adam epsilon | 1e-6 |
| Adam beta weights | 0.9, 0.98 |
| learning rate scheduler | Warmup linear |
| Weight decay | 0.01 |
| Warmup proportion | 0.06 |

Table 10: Hyperparameters for hard and soft prompt-based fine-tuning.

| Hyperparameter | Assignment |
| --- | --- |
| number of steps | 100 epochs |
| batch size | 256 |
| maximum learning rate | 1e-05, 1e-4 |
| learning rate optimizer | AdamW |
| Adam epsilon | 1e-6 |
| Adam beta weights | 0.9, 0.98 |
| learning rate scheduler | Warmup linear |
| Weight decay | 0.01 |
| Warmup proportion | 0.06 |
| Masking Probability | 0.15 |

Table 11: Hyperparameters for both conventional continued pre-training (TAPT) and prompt-based conventional fine-tuning (PCP).

| Hyperparameter | Assignment |
| --- | --- |
| number of steps | 12 800 or 25 600 steps |
| batch size | 16 |
| learning rate | 1e-05, 2e-05, 5e-05 |
| learning rate optimizer | AdamW |
| maximum sequence length | 256 |
| learning rate scheduler | Warmup linear |
| Warmup proportion | 0.05 |
| learning rate decay | linear |

Table 12: Hyperparameters for self training. Algorithm-specific hyperparameters will be released in configuration files with the code.

