# OpenReview forum: "Don’t Stop Pretraining? Make Prompt-based Fine-tuning Powerful Learner"
_NeurIPS.cc/2023/Conference — NeurIPS 2023 poster_

### Official Review · Reviewer_bM1j · 2023-06-29

**Soundness:** 3 good
**Presentation:** 4 excellent
**Contribution:** 3 good
**Rating:** 7
**Confidence:** 5

**Summary:**

The paper proposes a new approach to pre-training language models called Prompt-based Continued Pre-training (PCP). PCP combines the idea of instruction tuning with conventional continued pre-training. The authors argue that PCP can improve the performance of prompt-based fine-tuning on a variety of natural language processing tasks.

**Strengths:**

1. The paper presents a clear comparison of the proposed PCP with the conventional continued pre-training. The evidence on benchmark datasets is strong to support the claim that PCP performs better on these tasks.
2. The paper is well-written and easy to understand. The authors do a good job of explaining the technical details of the proposed approach, and they provide clear and concise summaries of the experimental results.
3. The approach is simple and easy to adopt.


**Weaknesses:**

1. It is not well understood why conventional continued pretraining (TAPT) is doing so poorly on sentence-pair tasks.
2. Because PCP makes use of templated prompts to align continued pretraining and fine-tuning, with additional pseudo-labels. It is unclear whether the gain is due to prompt alignment or the additional data augmentation from pseudo-labels. Further ablation is necessary.
3. It is unclear how well the proposed approach generalizes to other types of pretraining objectives such as language modeling.


**Questions:**

1. More ablations and analysis are needed to understand why TAPT is not working for sentence-pair tasks.
2. More experiments to ablate the effect of template alignment vs data augmentation via pseudo-labels.


**Limitations:**

Generalization of the approach to other pretraining objectives is unclear.

---

> ### Author Rebuttal · Authors · 2023-08-08
>
> We are grateful for the reviewer (bM1j)’s thoughtful and thorough evaluation of our paper. We are genuinely appreciative of the positive feedback concerning simplicity and effectiveness of our approach, solid experimental evidence, and the quality of our presentation. We would like to respond to the reviewer's valuable feedback as follows:
>
> ***Because PCP makes use of templated prompts to align continued pretraining and fine-tuning, with additional pseudo-labels. It is unclear whether the gain is due to prompt alignment or the additional data augmentation from pseudo-labels. Further ablation is necessary.***
>
> Thank you for your insightful comments and suggestions. To address this question, we conduct an additional ablation study, where we solely utilize pseudo labels or templates in our proposed method PCP. This ablation study is carried out using soft prompt-based fine-tuning. As shown in Table below, the experimental results indicate that using either labels or templates exclusively will hurt the model's performance. This highlights the vital importance of integrating both templates and pseudo labels into our proposed method.
>
> |          | SST-2 | SST-5 | MR | CR | MPQA | Subj | TREC | CoLA | Mean |
> |----------|-------|-------|----|----|------|------|------|------|------|
> |Prompt FT | 92.5 | 48.0 | 86.8 | 90.8 | 81.2 | 90.3 | 83.0 | 4.9 | 72.2 |
> |Prompt FT+PCP| 93.9 | 50.7 | 89.8 | 92.0 | 88.3 | 94.9 | 88.6 | 21.5 | 77.5 |
> |Prompt FT+PCP(Pseudo Labels Only)| 93.7 |50.8 | 87.7 | 91.3 | 85.1 | 94.3 | 85.7 | -0.7 | 73.5 |
> |Prompt FT+PCP(Template Only)| 90.7 | 43.5 | 88.6 | 92.6 | 82.0 | 95.1 | 84.1 | 0.7 | 72.2 |
>
> ***It is not well understood why conventional continued pretraining (TAPT) is doing so poorly on sentence-pair tasks?***
>
> We express our gratitude to the reviewer for their constructive suggestion. In response to this query, we have undertaken further experiments. For detailed information, please refer to our general response to all reviewers.
>
> ***It is unclear how well the proposed approach generalizes to other types of pretraining objectives such as language modeling.***
>
> We appreciate your insightful comments and questions regarding the generalizability of our proposed approach to other pre-training objectives. We recognize the value of exploring this avenue. We will do our best to get the necessary resources to investigate this important direction, and we leave this to future work.

---

> > ### Comment · Reviewer_bM1j · 2023-08-11
> >
> > I appreciate the due-diligence the authors took to address some of the concerns the reviewers including myself raised. I don't have further concerns despite that we still don't fully understand why TAPT performs poorly on sentence pair tasks (the authors did a very thorough ablation given the limited time).
> >
> > I would change my Rating to 7: Accept.

---

> > > ### Author Response · Authors · 2023-08-11
> > >
> > > We are very glad to hear that our response is helpful! May we kindly ask the reviewer to consider updating the score officially, as the organizers have now rectified the issue with updating scores? Thank you again for your understanding and assistance.

---

### Official Review · Reviewer_fyTk · 2023-07-03

**Soundness:** 3 good
**Presentation:** 2 fair
**Contribution:** 3 good
**Rating:** 6
**Confidence:** 4

**Summary:**


This work re-examines a well-known technique in the NLP literature, called continued pre-training, that can be utilized to enhance the performance of language models on downstream tasks (in this case, for classification and regression tasks).
The authors have revealed that the conventional approach to continued pre-training is ineffective when applied to sentence-pair tasks and prompt-based fine-tuning settings.
Based on their findings, the authors propose a simple method that involves training on the masking language modeling objective while augmenting the text input with prompts.
This proposed approach has demonstrated its effectiveness across a range of tasks and configurations, suggesting that it can be an effective choice for natural language understanding tasks when using Transformer encoder-based models.

**Strengths:**

- The proposed method is technically simple and effective.
- The work begins with a compelling finding that challenges the effectiveness of the conventional approach, which was previously regarded as effective.
- The authors made efforts to provide analysis from multiple perspectives, aiming to convince readers of the effectiveness of the proposed approach.



**Weaknesses:**

- Although the paper offers a detailed explanation and analysis, there are still ambiguous points that need to be clarified for a better understanding by the readers. Let me ask about this point in the following Questions section.
- There is room for improving the grammar and fluency of the paper's writing.
- I would like to see a thorough analysis of the factors that potentially contribute to the success of the proposed method. For instance, it would be valuable to explore the significance of including labels in continued fine-tuning (although partially considered in Section 4.4) and investigate why sentence-pair tasks do not benefit from the conventional TAPT approach.




**Questions:**


To the best of my understanding, it appears that your method incorporates prompts and their corresponding verbalized labels into the process of continued pre-training. If this is the case, there is a possibility that the tokens for verbalized labels are chosen as the target for masked language modeling. This implies that the method might be performing a similar task during both continued pre-training and fine-tuning. Consequently, there is a potential risk that the effectiveness of the proposed method is primarily attributed to unintentionally prolonged fine-tuning. If this turns out to be true, it could diminish the significance of this work.

To alleviate any ambiguity regarding this point, it would be helpful if you could provide concrete examples demonstrating how your method is applied to real data instances. This would offer a clearer understanding of the methodology and help evaluate its effectiveness more accurately.

**Limitations:**

The evaluation primarily focuses on the utilization of a single model, specifically RoBERTa-large, for relatively straightforward downstream tasks such as classification. These tasks are considered comparatively easier to solve.

---

> ### Author Rebuttal · Authors · 2023-08-08
>
> We express our gratitude to the reviewer (fyTk) for the insightful and comprehensive assessment. It is heartening to receive positive feedback on the simplicity and effectiveness of our approach, as well as the quality of our analysis. Furthermore, we appreciate the acknowledgement of our contribution in identifying the limitations of the previous approach. We would like to respond to the reviewer's valuable feedback as follows:
>
> ***I would like to see a thorough analysis of the factors that potentially contribute to the success of the proposed method. For instance, it would be valuable to explore the significance of including labels in continued fine-tuning (although partially considered in Section 4.4)***
>
> We are sincerely thankful for valuable comments and suggestions. As suggested by the reviewer, we carry out an additional experiment which involves the inclusion of labels in the Task-Adaptive Pre-training (TAPT) before performing CLS-based fine-tuning. The results of this experiment are presented in Table below. Our findings suggest that the inclusion of labels in the TAPT phase does not notably improve the model's performance, and there remains a considerable performance gap with our proposed method. We hope that our response has adequately addressed your initial concerns and would be most grateful if the reviewer could consider a score increase accordingly.
>
> |          | SST-2 | SST-5 | MR | CR | MPQA | Subj | TREC | CoLA | Mean |
> |----------|-------|-------|----|----|------|------|------|------|------|
> |CLS FT    | 81.2  | 41.7  | 76.3 | 79.5 | 65.1 | 91.7 | 80.3 | 26.7 | 67.1 |
> |CLS FT+TAPT| 88.2 | 43.4 | 86.1 | 86.2 | 73.7 | 94.2 | 80.4 | 1.9 | 69.3 |
> |CLS FT+TAPT with Labels| 89.2 | 42.3 | 84.2 | 85.7 | 75.8 | 93.8 | 84.5 | 0.7 | 69.5 |
> |Prompt FT+PCP| 93.9 | 50.7 | 89.8 | 92.0 | 88.3 | 94.9 | 88.6 | 21.5 | 77.5 |
>
>
> ***investigate why sentence-pair tasks do not benefit from the conventional TAPT approach.***
>
> We express our gratitude to the reviewer for their constructive suggestion. In response to this query, we have undertaken further experiments. For detailed information, please refer to our general response to all reviewers.
>
> ***To the best of my understanding, it appears that your method incorporates prompts and their corresponding verbalized labels into the process of continued pre-training. If this is the case, there is a possibility that the tokens for verbalized labels are chosen as the target for masked language modeling. This implies that the method might be performing a similar task during both continued pre-training and fine-tuning. Consequently, there is a potential risk that the effectiveness of the proposed method is primarily attributed to unintentionally prolonged fine-tuning. If this turns out to be true, it could diminish the significance of this work.***
>
> We appreciate the insightful comments and suggestions. To answer the reviewer’s question, we conduct additional experiments. We train cls-based fine-tuning 5 times more steps (5k steps in total) from the TAPT checkpoint. As shown in Table below, our results reveal that prolonged fine-tuning only brings about a marginal improvement of only 0.1% across the eight tasks. Notably, this still falls significantly short of our proposed method (8.1% in absolute). We will include this in the revised version of our paper.
>
> |          | SST-2 | SST-5 | MR | CR | MPQA | Subj | TREC | CoLA | Mean |
> |----------|-------|-------|----|----|------|------|------|------|------|
> |CLS FT(1k steps)+TAPT| 88.2 | 43.4 | 86.1 | 86.2 | 73.7 | 94.2 | 80.4 | 1.9 | 69.3 |
> |CLS FT(5k steps)+TAPT| 89.6 | 43.4 | 86.7 | 87.0 | 72.9 | 94.6 | 79.0 | 1.7 | 69.4 |
> |Prompt FT(1k steps)+PCP| 93.9 | 50.7 | 89.8 | 92.0 | 88.3 | 94.9 | 88.6 | 21.5 | 77.5 |
>
>
> ***The evaluation primarily focuses on the utilization of a single model, specifically RoBERTa-large, for relatively straightforward downstream tasks such as classification. These tasks are considered comparatively easier to solve.***
>
> We appreciate the insightful comments and suggestions. To facilitate a direct comparison, we conducted experiments utilizing benchmarks that are widely recognized and employed in previous research [1,2]. This approach aligns with previous studies that focus on prompt-based fine-tuning, a benchmarking methodology that is widely accepted within the field.
>
> [1] Tianyu Gao, Adam Fisch, and Danqi Chen. Making Pre-trained Language Models Better Few-shot Learners. ACL 2021.
>
> [2] Ningyu Zhang, Luoqiu Li, Xiang Chen, Shumin Deng, Zhen Bi, Chuanqi Tan, Fei Huang, and Huajun Chen. Differentiable prompt makes pre-trained language models better few-shot learners. ICLR 2022.
>
> ***There is room for improving the grammar and fluency of the paper's writing.***
>
> We are thankful to the reviewer for their constructive critique. We will conduct a thorough review and improve the writing in our revised version.

---

### Official Review · Reviewer_pe8v · 2023-07-04

**Soundness:** 3 good
**Presentation:** 3 good
**Contribution:** 3 good
**Rating:** 7
**Confidence:** 3

**Summary:**

This paper explores the problem of continued pre-training on task-related text. The authors discovered that conventional continued pre-training methods may not be very effective and can even have a negative impact on fine-tuning performance. To address this, they introduce prompt-based continued pre-training. The approach involves generating pseudo labels on unlabeled data using a fine-tuned model and constructing a prompt-based pre-training corpus by applying templates to the pseudo labeled dataset. The researchers then utilize this corpus, which incorporates prompt information, for continued pre-training and prompt-based fine-tuning. Experimental evaluations conducted on various datasets validate the effectiveness of the proposed approach, as it achieves significant improvements over the baseline methods.

**Strengths:**

* The researchers identify the limitations of conventional continued pre-training on task-related text and propose an innovative solution.
* The experimental results provide strong evidence for the effectiveness of the proposed approach.

**Weaknesses:**

No major weaknesses have been identified in this paper.

**Questions:**

Is it possible to extend this approach to the task of language modeling fine-tuning?

---

> ### Author Rebuttal · Authors · 2023-08-08
>
> We appreciate the effort and time by the reviewer (pe8v). We are heartened by their positive appraisal of our work and their recognition that no major weaknesses have been identified in the paper. We would like to respond to their invaluable feedback as follows:
>
> ***Is it possible to extend this approach to the task of language modeling fine-tuning?***
>
> We greatly appreciate the reviewer's thoughtful comments and suggestions. If we understand correctly, the idea of incorporating prompt-based language modeling (prompt-based continued pre-training) as an auxiliary loss during fine-tuning is indeed a compelling one. We will definitely consider this perspective, viewing it as a potential route for exploration in our upcoming research.

---

### Official Review · Reviewer_TYSB · 2023-07-07

**Soundness:** 3 good
**Presentation:** 3 good
**Contribution:** 3 good
**Rating:** 5
**Confidence:** 4

**Summary:**

This paper makes a contribution by studying how to adapt pre-trained models to downstream tasks. The authors identify the limitations of TAPTs and show when they do not work well. They then propose PCP, a better algorithm that can adapt a pre-trained model to a target task. PCP is shown to be more effective than TAPTs on a variety of tasks.


**Strengths:**

The paper provides an intriguing analysis of when TAPT style is effective for fine-tuning. TAPT is not very effective on sentence-pair tasks or prompt-based fine-tuning, particularly.

**Weaknesses:**

The paper does a good job of pointing out the weaknesses of TAPT, but it is less clear why PCP works better. The authors do not need to know exactly why it works, but some hypotheses would be helpful. For example, why does PCP-style pretraining work better for prompt-based fine-tuning? Is it because the "pretraining" is more similar to "fine-tuning" in PCP? If so, how can we understand why PCP also works for sentence pair tasks?

The presentation of the paper could also be improved. Figure 2 is not very easy to understand, and the overall flow of the paper is not as clear as it could be.

Overall, the paper could be improved by providing more clarity and explanation.

**Questions:**

Cloud the authors provide more intuitions or analysis why PCP works better? Based on the intuitions, what kind of additional analysis can be done here?


**Limitations:**

The paper only applies on text classifications tasks, ignoring many text generation tasks.

---

> ### Author Rebuttal · Authors · 2023-08-08
>
> We are grateful for the reviewer (TYSB)’s thoughtful and thorough evaluation of our paper. We sincerely appreciate the positive feedback regarding our intriguing analysis and contribution to identify the limitations of the previous approach. We would like to address the reviewer's valuable feedback as follows:
>
> ***The paper does a good job of pointing out the weaknesses of TAPT, but it is less clear why PCP works better. The authors do not need to know exactly why it works, but some hypotheses would be helpful. For example, why does PCP-style pretraining work better for prompt-based fine-tuning? Is it because the "pretraining" is more similar to "fine-tuning" in PCP? If so, how can we understand why PCP also works for sentence pair tasks?***
>
> We acknowledge the reviewer's inquiry regarding the intuition of our method. The intuition of our method is that `language model will benefit from seeing the template/prompt before the task, no matter using supervised learning or unsupervised learning objectives`:
> - As we mentioned in the paper, the importance of presenting the templates to the language models have been also shown in various instruction tuning works. Our work differs from these works in two main aspects: (1) they aim to improve the zero-shot learning performance, while our work aims to improve the fine-tuning performance; and (2) they learn the template knowledge through supervised learning objectives, while we use unsupervised learning. However, the intuition (why PCP works better) is similar: showing the templates/prompt to the language models can improve the performance on the downstream tasks.
> - To explain why PCP works, we attribute the reason that PCP works better than TAPT to *presenting the prompt template to the model* as we mentioned in the paper, because this is the only difference between our method and TAPT. We find that TAPT only tells the model about how the text for the target task looks like (let us call it in-domain knowledge). In contrast, our proposed method PCP tells the model not only the in-domain knowledge but also the prompt information that will be used for fine-tuning on the target task. We also agree with the reviewer that the PCP brings the continued pre-training closer to the prompt-based fine-tuning. We ablate the importance of templates and labels in the PCP in our subsequent response to the reviewer.
> We will clarify this in the revised version of our paper.
>
> ***Cloud the authors provide more intuitions or analysis why PCP works better? Based on these intuitions, what kind of additional analysis can be done here?***
>
> The main intuition behind our PCP is that it is important to show the model with the template/prompt that will be used in the target task. We perform an ablation study to emphasize the importance of including both the template/prompt and label in PCP. As presented in Table below, the experimental results suggest that relying solely on either labels or templates will hurt the model's performance. This highlights the importance of integrating both templates and pseudo labels into our proposed method PCP.
>
> |          | SST-2 | SST-5 | MR | CR | MPQA | Subj | TREC | CoLA | Mean |
> |----------|-------|-------|----|----|------|------|------|------|------|
> |Prompt FT | 92.5 | 48.0 | 86.8 | 90.8 | 81.2 | 90.3 | 83.0 | 4.9 | 72.2 |
> |Prompt FT+PCP| 93.9 | 50.7 | 89.8 | 92.0 | 88.3 | 94.9 | 88.6 | 21.5 | 77.5 |
> |Prompt FT+PCP(Pseudo Labels Only)| 93.7 |50.8 | 87.7 | 91.3 | 85.1 | 94.3 | 85.7 | -0.7 | 73.5 |
> |Prompt FT+PCP(Template Only)| 90.7 | 43.5 | 88.6 | 92.6 | 82.0 | 95.1 | 84.1 | 0.7 | 72.2 |
>
> ***The presentation of the paper could also be improved. Figure 2 is not very easy to understand, and the overall flow of the paper is not as clear as it could be. Overall, the paper could be improved by providing more clarity and explanation.***
>
> We express our gratitude to the reviewer for their insightful feedback on the presentation of our paper. We are committed to improving this aspect in our revised version. The presentation of this paper will be improved as follows:
> - We will merge Figure 2 with the aforementioned intuition.
> - To further clarify our work with more explanation, all five additional experiments mentioned in our rebuttal will be incorporated into the revised version of our paper.

---

### Official Review · Reviewer_CwVD · 2023-07-09

**Soundness:** 3 good
**Presentation:** 3 good
**Contribution:** 3 good
**Rating:** 5
**Confidence:** 4

**Summary:**

The proposed method (CPC) is built on top of pseudo-labeling and continued pre-training via masked language modeling. This method provides an alternative way to use pseudo-labeled data before fine-tuning the model on downstream tasks. It improves the TAPT method and other semi-supervised methods for text classification.

**Strengths:**

- The novelty of this paper lies in using pseudo-labeled data for masked language modeling. This idea is simple, yet effective. Even though pseudo-labeling and continued MLM training are not new, the way to combine these two is new.
- Extensive experiments on a large number of text classification tasks show the effectiveness of the method.


**Weaknesses:**

- One of the motivations (mentioned in the introduction) is that TAPT does not work well on sentence pair classification. However, it’s still unclear why TAPT does not work well on sentence pair classification while CPC can address this issue.
- Although CPC works empirically better than TAPT, the intuition of CPC is still unclear. Why should we continue pre-training on pseudo-labeled data instead of unlabeled data in TAPT?


**Questions:**

1.	What masking strategy is used in Step 2? Are you using the random masking strategy as used in Robert?
2.	If you have pseudo labels in CPC, is random masking in Step 2 the optimal strategy? How about some selective masking strategies (e.g., PMI-masking [1])?
3.	The paper is missing a self-training baseline, where you only mask the pseudo labels for MLM pre-training in Step 2.
4.	It’d be good to include the scores of TAPT in Fig 3, representing “no label + FT”. It will be helpful to compare “wrong label + FT” and “no label + FT”. Because the method is sensitive to the pseudo-labeling performance, it'd be good to add a few-shot setting where the base model is fine-tuned on few-shot examples in Step 1. And compare CPC with TAPT in the few-shot setup.
5.	To better compare TAPT (“no label +FT”) with CPC (“pseudo-label +FT”), we should have a better understanding of the impact of the “pseudo-label” in the MLM process. Does the addition of “pseudo-label” in the input sentence improve the MLM prediction accuracy?

[1] PMI-Masking: Principled masking of correlated spans

---

> ### Author Rebuttal · Authors · 2023-08-08
>
> We appreciate the effort and time by the reviewer (CwVD). We are thrilled to receive positive feedback on the novelty and simplicity of our method, along with the extensive experiments that underscores the effectiveness of our method. We would like to address the reviewer's valuable feedback as follows:
>
> ***it’s still unclear why TAPT does not work well on sentence pair classification while CPC can address this issue.***
>
> We acknowledge the reviewer's valid question regarding the performance of TAPT on sentence pair tasks. In response, we have performed additional experiments to answer this question. We would invite the reviewer to refer to our general response for details.
>
> ***Although CPC works empirically better than TAPT, the intuition of CPC is still unclear. Why should we continue pre-training on pseudo-labeled data instead of unlabeled data in TAPT?***
>
> We appreciate the reviewer's question. As mentioned in our paper, our intuition for PCP is that `language model will benefit from seeing the template/prompt before the target task`, as the importance of presenting the templates to the language models have been shown in various instruction tuning works. We find that TAPT only tells the model about how the text for the target task looks like (let us call it in-domain knowledge). Based on this finding and intuition, we propose PCP, which not only tells the model the in-domain knowledge, but also the prompt information that will be used for fine-tuning on the target task. This also makes the objective of continued pre-training closer to the objective of fine-tuning. We will make it clear in our revised version.
>
> ***What masking strategy is used in Step 2? Are you using the random masking strategy as used in Robert?***
>
> Yes, we use the same masking strategy as used in RoBERTa. We dynamically mask 15% tokens. We will make this clear in our revised version.
>
> ***If you have pseudo labels in CPC, is random masking in Step 2 the optimal strategy? How about some selective masking strategies (e.g., PMI-masking [1])***
>
> We appreciate the reviewer's suggestion about PMI-masking, which is indeed highly relevant to our work. We will discuss this work in our revised version. We agree with the reviewer that there exists some potential variants, such as using PMI-Masking. However, it's essential to highlight that such a method could be applicable to both TAPT and our proposed method PCP. Our research is primarily centered on contrasting TAPT and PCP, rather than exploring strategies that could be equally beneficial for both. Moreover, assessing these variants across a broad range of datasets imposes a large resource burden on us. Consequently, we plan to leave the evaluation of these variants for future work.
>
> ***The paper is missing a self-training baseline, where you only mask the pseudo labels for MLM pre-training in Step 2***
>
> We thank the reviewer for their constructive feedback. To clarify, when we exclusively mask the pseudo labels, our task reverts to prompt-based fine-tuning. We compare our proposed method with 4 state-of-the-art self-training models in Table 2 of our paper, where all these self-training baselines use the prompt-based fine-tuning as the backbone. Results show that PCP outperforms these state-of-the-art self-training models. We will clarify this in our revised version.
>
> ***It’d be good to include the scores of TAPT in Fig 3, representing “no label + FT”. It will be helpful to compare “wrong label + FT” and “no label + FT”.***
>
> We thank the reviewer for this valuable suggestion. We will definitely do this.
>
> ***Because the method is sensitive to the pseudo-labeling performance, it'd be good to add a few-shot setting where the base model is fine-tuned on few-shot examples in Step 1. And compare CPC with TAPT in the few-shot setup.***
>
> We thank the reviewer for this valuable suggestion. Actually, we exactly follow the few-shot learning setting in the prior work [1], where the only difference is that we use additional unlabelled data. The pseudo-labels for this unlabelled data in PCP are assigned by the model trained in the few-shot learning setting. As shown in Figure 1 of our paper, PCP brings substantial improvement when using a base model trained on few-shot examples to produce pseudo-label. We will clarify this in our revised version.
>
> [1] Tianyu Gao, Adam Fisch, and Danqi Chen. Making Pre-trained Language Models Better Few-shot Learners. ACL 2021.
>
> ***To better compare TAPT (“no label +FT”) with CPC (“pseudo-label +FT”), we should have a better understanding of the impact of the “pseudo-label” in the MLM process. Does the addition of “pseudo-label” in the input sentence improve the MLM prediction accuracy?***
>
> We thank the reviewer for the insightful suggestion. In response, we have performed additional experiments, as shown in Table below. Our results indicate that PCP indeed improves the accuracy of MLM. In terms of average accuracy across 8 single-sentence tasks, TAPT attains a score of 0.6713, while PCP obtains a higher average accuracy of 0.7225. We also find that the accuracy of the pseudo label does not appear to significantly influence the MLM accuracy. To be specific, PCP with correct labels yields an average accuracy of 0.7188, while PCP with wrong labels records an average accuracy of 0.7238. Similar results can be observed for sentence pair tasks. The reason for improved accuracy might benefit from the usage of the same template in all sentences, which is easier to predict. We will include this in our revised version. We hope that our response has adequately addressed the reviewer’s concerns and would be most grateful if the reviewer could consider a score increase accordingly.
>
>
> |          |    TAPT   | PCP     | PCP (correct-label) | PCP (wrong-label) |
> | ----- | -----   | ---- |---               | ----   |
> | Single Sentence Tasks |  0.6713|   0.7225  |    0.7188 |  0.7238 |
> | Sentence Pair Tasks | 0.7586 | 0.7700  |  0.7800 |  0.7728 |

---

### Author Rebuttal · Authors · 2023-08-08

We appreicate all the reviewers for dedicating their time and effort to evaluate our work.  We are thrilled to receive positive feedback on **the novelty of our approach** (CwVD,pe8v), **the simplicity and effectiveness of our approach** (CwVD,fyTk,bM1j), **solid experimental evidence** (CwVD,pe8v,bM1j), **intriguing/thorough analysis** (TYSB,fyTk), and **the quality of the presentation** (bM1j). We also thank the reviewer (TYSB,pe8v,fyTk) for acknowledging our **contribution to identify the limitations of the previous approach**.

To answer the reviewers’s questions, we conduct 5 additional experiments, regarding **the potential reason for why TAPT does not work on sentence pair tasks**, **the impact of PCP on the MLM accuracy**, **an ablation study on template only and label only**, **the impact of adding label to TAPT**, **the impact of training CLS-based fine-tuning longer**. We hope that our response, paired with these additional experiments, will address the reviewers' concerns.

One common question is about why TAPT does not work on sentence pair tasks. We delve into this particular issue below. For the remaining concerns, we will respond to each reviewer individually. We have evaluated three possible explanations for TAPT's ineffectiveness on sentence pair tasks: **dataset size**, **sentence pairs with higher similarity than what was observed in pre-training data**, and **lack of separation within sentence pairs**. Our experimental results suggest that the ineffectiveness of TAPT on the sentence pair tasks is not an isolated incident but a recurring issue. Below we discuss each setting in detail.
- **Training data size**. We wonder if the size of training data could be a limiting factor. To test this, we perform TAPT on MNLI, MNLI-mm, SNLI, QNLI, QQP datasets, with up to 360k training data. Our experimental results, detailed in our paper's Appendix, reveal that training TAPT with a large corpus still undermines the performance of cls-based fine-tuning on sentence pair tasks.
- **High similarity within sentence pairs**. We consider that the high similarity between the sentence pairs might conflict with the word distribution that the model has observed during model pre-training. For instance, in the MNLI task, two sentences are `Salt kept the town fed` and `Salt kept the town thriving`. To explore this, we perform TAPT on two different settings, one where we continually pre-train TAPT on randomly paired sentences within the dataset and another where we continually pre-train TAPT using just the first sentence of each pair. As shown in the Table below, the experimental results show that training TAPT with either case leads to even worse performance.
- **Token-based separation of sentence pairs**. In an attempt to mitigate the effect above, we also consider that distinguishing two sentences using distinct tokens might make a difference. To test this, we perform TAPT with two types of separate tokens, the special token from the tokenizer and the template used in PCP (without labels). As shown in the Table below, training TAPT with separate tokens between two sentences can somewhat mitigate the performance drop for cls-based fine-tuning on the sentence pair tasks. However, the results remain inferior compared to cls-based fine-tuning without the use of TAPT.

In conclusion, our investigations highlight the difficulties that TAPT faces on sentence pair tasks, while our proposed method PCP provides a simple yet effective solution. We hypothesize that TAPT's ineffectiveness for CLS-based fine-tuning on sentence pair tasks might be due to various factors, which we leave for a more comprehensive investigation in future work.


|           | MNLI      |MNLI-mm|SNLI |QNLI | RTE  | MRPC | QQP  |STS-B | Mean |
|---------- |-------    |-------|-----|-----|------|------|------|------|------|
|CLS FT     | 46.2      | 48.5  | 45.6| 61.4| 54.2 | 73.2 | 58.5 | 46.0 | 54.2 |
| +TAPT| 36.0      | 36.3  | 45.7| 55.6| 53.4 | 67.7 | 55.0 | 48.1 | 49.7 |
|+TAPT (Tokenizer Sep)| 36.4  | 37.5  | 50.5| 58.8| 50.8 | 63.5 | 59.2 | 48.8 | 50.7  |
|+TAPT (PCP Sep)| 36.3  | 36.7  | 64.6| 58.3| 51.2 | 65.3 | 57.4 | 44.5 | 51.8  |
|+TAPT(random sent pair)| 34.8 | 35.4  | 37.7| 52.2| 51.2 | 64.8 | 56.9 | 23.8  | 44.6  |
|+TAPT(first sent only)| 35.6 | 35.9  | 42.7| 52.2| 52.6 | 62.5 | 53.6 | 16.7 | 44.0 |

---

### Decision · Program_Chairs · 2023-09-21

**Decision:**

Accept (poster)

**Comment:**

This paper argues that the conventional continued pre-training method may not always bring improvements for downstream tasks under certain scenarios. The authors then propose a novel method called "Prompt-based Continued Pre-training" which is shown to be able to improve the performance on downstream tasks. All the reviewers are largely positive on this work, with some main concerns such as the reasons the approach works better than previous approaches, and why the previous approach (TAPT) performs poorly on the sentence-pair classification task. Though the latter remains largely unanswered in this work, the current paper contains sufficient new knowledge and methodologies that are worth sharing with the community.